**Understanding model spread in sea ice volume by attribution of model differences in seasonal ice growth and melt**

Alex E. West[1], Edward W. Blockley[1], Matthew Collins[2]

[1]Met Office Hadley Centre, FitzRoy Road, Exeter EX1 3PB

[2]Centre for Engineering, Mathematics and Physical Sciences, University of Exeter, Stocker Road, Exeter EX4 4PY

*Correspondence to*: Alex E. West (alex.west@metoffice.gov.uk)

**Abstract.** Arctic sea ice is declining rapidly, but predictions of its future loss are made difficult by the large spread both in present-day and in future sea ice area and volume; hence, there is a need to better understand the drivers of model spread in sea ice state. Here we present a framework for understanding differences between modelled sea ice simulations based on attributing seasonal ice growth and melt differences. In the method presented, the net downward surface flux is treated as the

principal driver of seasonal sea ice growth and melt. An energy balance approach is used to estimate the pointwise effect of model differences in key Arctic climate variables on this surface flux, and hence on seasonal sea ice growth and melt. We compare three models with very different historical sea ice simulations: HadGEM2-ES, HadGEM3-GC3.1 and UKESM1.0. The largest driver of differences in ice growth / melt between these models is shown to be the ice area in summer (representing the surface albedo feedback) and the ice thickness distribution in winter (the thickness-growth feedback).

Differences in snow and melt-pond cover during the early summer exert a smaller effect on the seasonal growth and melt, hence representing the drivers of model differences in both this and in the sea ice volume. In particular, the direct impacts on sea ice growth / melt of differing model parameterisations of snow area and of melt-ponds are shown to be small but non-negligible.

## 1 Introduction

Arctic sea ice has undergone dramatic changes in recent decades, with a decline of 0.88 x 10[6] km2 per decade in September extent observed from 1979-2020 according to the HadISST.2.2 dataset (Titchner and Rayner, 2014) and associated thinning in summer and winter (Lindsay and Schweiger, 2015). Associated with the decline in sea ice has been winter warming (e.g. Graham et al., 2017), earlier onset of melt over the ice (Markus et al., 2009), and later onset of freezing (Stammerjohn et al., 2012).

Sea ice is an important component of the climate system, due to its high albedo, its ability to insulate the atmosphere from oceanic heat during the winter, and its effects on ocean circulation. Hence the accurate modelling, and future prediction of sea ice are of primary importance for climate science. These issues are an important focus of the Coupled Model

Intercomparison Project (CMIP), in which results from coupled models worldwide are collated, compared and evaluated in a series of 'phases' every few years.

In the most recent three phases of CMIP (CMIP3, CMIP5 and CMIP6), sea ice was evaluated in the present-day by Stroeve et al. (2007), Stroeve et al. (2012), Shu et al. (2015) and Notz et al. (2020) amongst others, finding substantial spread in ice extent and volume. All studies have found a tendency to underestimate speed of sea ice extent loss; this may be due to internal variability (e.g. Notz, 2015), although Rosenblum and Eisenman (2017) and Notz et al. (2020) found that speed of decline was also underestimated as a function of rate of global temperature change.

There have been multiple attempts to understand the reasons behind the large spread in model sea ice simulation. Holland (2008) showed that model spread in Arctic sea ice volume during the historical period explained a large proportion of model spread in the rate of Arctic sea ice loss. Eisenman et al. (2007) and DeWeaver et al. (2008) linked differences in sea ice simulation in the CMIP3 ensemble to differences in atmospheric forcing using idealised models but disagreed on the exact relationship. Massonnet et al. (2012) evaluated a variety of metrics of sea ice state in CMIP5, finding a number of statistical

relationships between present and future sea ice state, but did not examine the underlying drivers in detail. Boeke and Taylor (2016) evaluated Arctic surface radiation in the CMIP5 ensemble, showing upwelling SW to be strongly dependent on sea ice area, but did not examine causation in the opposite direction. Keen et al. (2021) compared mass budget terms across the CMIP6 ensemble, identifying links between sea ice model parameters and the relative size of individual terms. In this study we aim to build on the strengths of these previous approaches, by combining evaluation of Arctic climate variables with

simple physical relationships between these variables.

The framework is based upon the induced surface flux (ISF) bias method of West et al. (2019), which is extended and enhanced to allow comparison of models to each other (as opposed to comparing models only to observations of the real world). The net downward surface flux is treated as the principal driver of the sea ice growth and melt. We use a system of simple models to understand the ways in which differences in individual model variables drive differences in the surface

flux, and hence in the sea ice growth and melt. In particular, we use the framework to separate the effects of model differences in sea ice area and thickness on sea ice growth and melt (representing the two most important feedbacks of the sea ice state, the surface albedo feedback and thickness-growth feedback) from the effects of model differences in other Arctic climate variables (representing, in a sense, external forcing of the Arctic sea ice state). In this way, we can analyse how model differences in forcing variables drive differences in ice state, and ice growth and melt, as a coupled system.

The framework is similar to that used by Holland and Landrum (2015) to quantify the contribution of changes in surface albedo and in downwelling SW to changes in net SW over 3 periods of the 21st century, but differs in two ways. Firstly, it quantifies the contribution of model processes to model biases, and inter-model differences, rather than to changes in time within model runs. Secondly, it is effectively a generalisation of this method, as it quantifies the contributions of model differences in a larger number of model variables.

The study is set out in the following way. In Section 2, the models and reference datasets used for evaluation are described. In Section 3, the Arctic climate simulations of the models are compared, and evaluated with respect to observations. In Section 4, the ISF framework is introduced and used to separate the effects of different surface albedo drivers on ice volume balance, and the effects of the thickness-growth feedback during winter from other variables. In Section 5, we discuss how the results enable understanding of the sea ice state / sea ice growth and melt as a coupled system. In Section 6, conclusions

are presented.

## 2 Models and reference data

### 2.1 Models

In this paper we evaluate the UK CMIP6 models HadGEM3-GC3.1 (Williams et al., 2017) and UKESM1.0 (Sellar et al., 2019), comparing them to the previous generation CMIP5 model HadGEM2-ES (Collins et al., 2011). All are fully coupled

models with interactive sea ice components, and HadGEM2-ES and UKESM1.0 employ additional components to simulate terrestrial and oceanic ecosystems, and tropospheric chemistry (see Collins et al., 2011; Sellar et al., 2019).

Within CMIP6 HadGEM3-GC3.1 was run at multiple resolutions: in this study, we evaluate the low-resolution configuration, HadGEM3-GC3.1-LL (Kuhlbrodt et al., 2018), for consistency with UKESM1.0 -LL which was run at low resolution only. In the LL configuration , the atmosphere model is integrated on the N96 grid, with a resolution of 1.25

degrees latitude and 1.07 degrees longitude. The ocean and sea ice models are integrated on the ORCA1 grid, an irregular grid with a resolution of approximately 50km in the Arctic. HadGEM2-ES was run at only one resolution, with the atmosphere model also on the N96 grid, and the ocean model on the HadGOM grid, a regular latitude-longitude grid with a resolution of one degree in polar regions.

We briefly describe differences in the sea ice components of the models here (differences in the atmospheric and ocean

components are described in more detail in Williams et al., 2018). Some features of the sea ice components are shared between all models. All employ a sub-grid-scale ice thickness distribution (Thorndike et al., 1975); elastic-viscous-plastic rheology (Hunke and Dukowicz, 1997) and incremental remapping (Lipscomb and Hunke, 2004). In addition, all models share a thermodynamic framework in which the surface energy balance over ice is calculated in the atmosphere model (West et al., 2016).

However, there are some fundamental differences between the sea ice component of HadGEM2-ES, which uses the native sea ice model of the HadGOM ocean model, and those of HadGEM3-GC3.1-LL and UKESM1.0-LL (henceforth referred to in this study as the CMIP6 models), which use the Los Alamos sea ice model CICE, version 5.1.2, in the GSI8 configuration, Ridley et al. 2018. Firstly, in HadGEM2-ES the sea ice has no heat capacity and responds instantly to surface thermodynamic forcing (zero-layer framework, appendix to Semtner (1979)), with vertical conduction assumed to be

uniform through the ice. In the CMIP6 models the sea ice is divided into 4 equally spaced layers, each with a heat capacity, with temperatures calculated according to surface forcing using a forwards-implicit scheme (multi-layer framework, Bitz and Lipscomb, 1999).

Secondly, the radiative effect of melt-ponds is modelled explicitly in the two newer models, using the topographic scheme of Flocco et al. (2014). In this scheme, while melt-pond albedo is a fixed parameter, melt-pond fraction of ice is able to vary according to the total volume of meltwater, and the ice topography. In HadGEM2-ES the radiative effect of melt-ponds is modelled implicitly, by reducing albedo linearly (from 0.84 to 0.67 for snow; from 0.64 to 0.565 for bare ice) as surface temperature increases from -1 to 0°C.

Finally, the sea ice model of HadGEM2-ES runs on a regular latitude-longitude grid, with a polar island, while those of HadGEM3-GC3.1 and UKESM1.0 run on the extended ORCA1 grid, an irregular tri-polar grid with poles in Antarctica, Russia and Canada.

**2.2 Reference data**

The reference datasets used to evaluate HadGEM3-GC3.1 and UKESM1.0 in section 3 are identical to those used in West et al. (2019) for HadGEM2-ES. A full description of these datasets is given in the earlier study, but they are summarised here, together with some probable biases identified.

For ice concentration, we use the NSIDC 'Sea Ice Concentrations from Nimbus-7 SMMR and DMSP SSM/I-SSMIS Passive Microwave Data, Version 1' (Cavalieri et al., 1996), as well as versions 1.2 and .2.2 of the Hadley Centre Ice and Sea Surface Temperature dataset (HadISST; Rayner et al., 2003 for HadISST1.2 and Titchner and Rayner, 2013 for HadISST.2.2). All datasets use satellite passive microwave imagery as their primary data source (SSMI), but the algorithm used to calculate sea ice area from microwave retrievals differs between the datasets. Uncertainty in ice area is known to be particularly high in the summer, as satellites cannot distinguish between meltponds and open water.

For ice thickness, we use the Pan-Arctic Ocean-Ice Modeling and Assimilation System (PIOMAS; Schweiger et al., 2011), an ice-ocean model forced by NCEP atmospheric data that assimilates ice concentration from HadISST and NSIDC. PIOMAS has been evaluated with respect to satellite observations (Laxon et al., 2013; Wang et al., 2016) and found to compare quite well, though Wang et al. (2016) found PIOMAS to underestimate winter ice thickness relative to ICESat by 0.31m over the Arctic Ocean from 2003-2008. We also use EnviSat radar altimetry observations from 1993-1999 (Laxon et al., 2003) and a regression analysis of submarine sonar observations from 1980-1999 (Rothrock et al., 2008). Because the latter two datasets are available only for restricted regions, they are used for model evaluation only and not for the ISF analysis.

For surface radiation, we use three datasets: CERES-EBAF (Loeb et al., 2009) and ISCCP-FD (Zhang et al., 2004) satellite observations, obtained by the application of radiative transfer algorithms to reflected SW and upwelling LW retrievals, and the ERAI reanalysis dataset (Dee et al., 2011). Comparison of downwelling LW observations from these datasets to in-situ observations by Lindsay et al (1998) suggested a high bias in ISCCP-FD but no clear bias in CERES-EBAF or ERAI. By contrast Lindsay et al. (2014) found ERAI to overestimate downwelling LW in winter by 15 $Wm^{-2}$ and underestimate downwelling SW in spring by 20 $Wm^{-2}$, but with no clear biases in other seasons, and Christenssen et al. (2016) found CERES-EBAF to underestimate downwelling LW radiation in winter by 10-15 $Wm^{-2}$ relative to in-situ observations at Point Barrow, Alaska.

Finally, for surface melt onset we use the NSIDC 'Snow Melt Onset Over Arctic Sea Ice from SMMR and SSM/I-SSMIS Brightness Temperatures' dataset (Anderson and Drobot, 2012).

## 3 Evaluating the UK CMIP6 models

In this section, we evaluate and compare the three models UKESM1.0-LL, HadGEM3-GC3.1-LL and HadGEM2-ES. We analyse first the sea ice state (ice area and volume), before proceeding to examine variables affecting the surface energy balance.

### 3.1 Sea ice state

Evaluation is performed for the 1980-1999 period, and for the Arctic Ocean region, defined in Figure 1b, unless otherwise stated. The evaluation is performed for the four historically-forced members of HadGEM3-GC3.1-LL and HadGEM2-ES, and the first six historically-forced members of UKESM1.0-LL. For energy fluxes the sign convention used is that downwards is positive.

Total Northern Hemisphere sea ice extent is higher in both UKESM1.0-LL and HadGEM3-GC3.1-LL than in HadGEM2-ES, particularly during the summer (Figure 1a). HadGEM2-ES tended to underestimate Arctic sea ice extent from June-October with respect to HadISST1.2, HadISST.2 and NSIDC, whereas ice extent in HadGEM3-GC3.1-LL and UKESM1.0-LL mainly lies within observational spread. However, HadGEM3-GC3.1-LL and UKESM1.0-LL simulate the minimum of the annual cycle in extent to occur in August, whereas in reality it occurs in September.

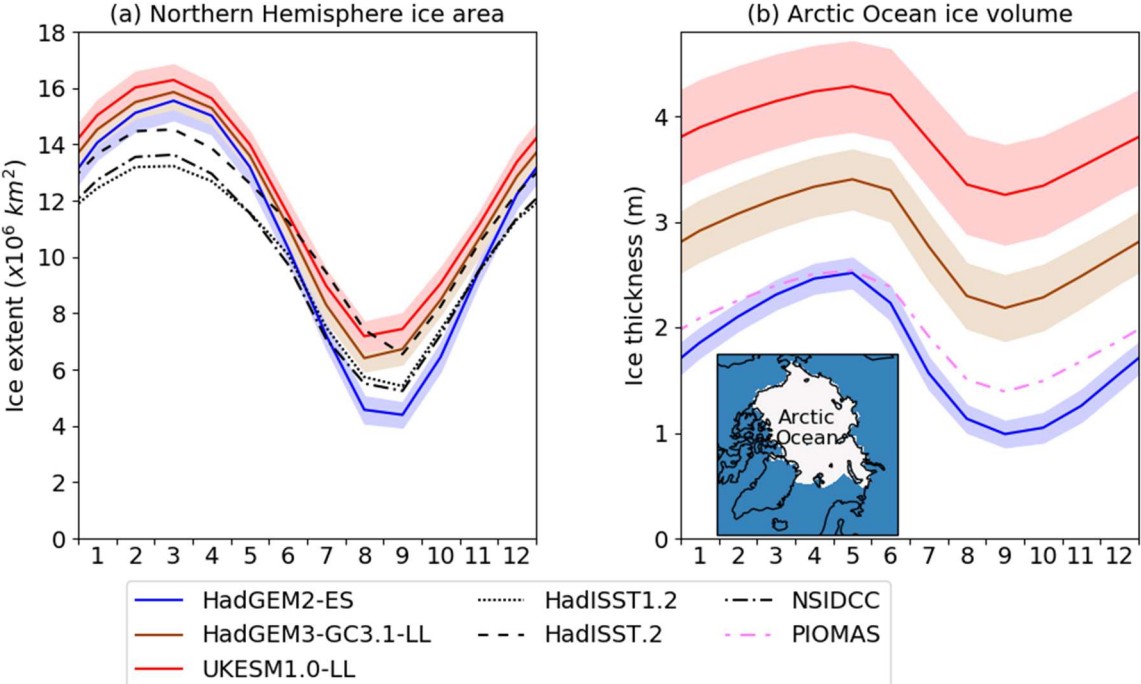

**Figure 1. (a) 1980-1999 average sea ice extent for the whole Northern Hemisphere, for the evaluated models and reference datasets (HadISST1.2, HadISST.2 and NSIDC); (b) 1980-1999 average sea ice volume over the Arctic Ocean region shown in the subpanel, for the evaluated models and PIOMAS. For each model, ensemble mean (lines) and twice standard deviation (shading) is shown.**

Sea ice thickness is greatest in UKESM1.0-LL (Figure 1b, annual mean Arctic Ocean region thickness 3.9m) and least in HadGEM2-ES (1.7m) throughout the year with HadGEM3-GC3.1-LL tending to lie between the two (2.8m). UKESM1.0-LL and HadGEM3-GC3.1-LL are biased high relative to the ice-ocean model PIOMAS (Figure 1b, annual mean thickness 1.9m), whereas HadGEM2-ES is biased low during the summer. We also evaluate the three models with respect to ERS satellite measurements from 1993-1999 and to the multiple regression of submarine data from 1980-1999 (not shown). In both cases UKESM1.0-LL and HadGEM3-GC3.1-LL are biased high and HadGEM2-ES is biased low year-round. However HadGEM3-GC3.1-LL lies somewhat closer to observations, and HadGEM2-ES somewhat further, than is the case when PIOMAS is reference dataset, consistent with PIOMAS being biased low during winter (Section 2.2). For all three models, internal variability (shown as twice the ensemble standard deviation) is low compared to the inter-model differences and the model biases.

The large differences in ice thickness between the models are associated also with differences in the amplitude of the seasonal cycle of ice thickness, hereafter referred to as ice growth / melt. Ice growth / melt is greatest in HadGEM2-ES

(1980-1999 Arctic Ocean mean 1.53m); it is lower in HadGEM3-GC3.1 (1.22m) and lower still in UKESM1.0-LL (1.03m). By comparison, PIOMAS annual ice growth / melt is similar to HadGEM3-GC3.1 at 1.15m. The correspondence between the model annual mean ice thicknesses and volume balances are broadly consistent with what would be expected from the ice thickness-growth feedback (Bitz and Roe, 2004): thicker ice tends to grow less in winter, because it conducts heat from the ocean to the atmosphere less efficiently. It is also consistent with the surface albedo feedback (Bitz, 2008): thinner ice

warms more quickly in early spring, creating melt-ponds sooner, and supports the creation of more extensive open water areas, leading to greater absorption of shortwave (SW) radiation and greater ice melt.

### 3.2 A conceptual picture of the proximate causes of ice melt/growth

To motivate the next stage of the model evaluation, and the ensuing analysis, we use an energy balance approach. The ice heat uptake arises as the sum of the heat flux from the atmosphere, from the ocean, and from outside the Arctic (Figure 2a),

and is itself composed of latent heat uptake (representing ice melt/growth) and sensible heat uptake. For the remainder of this study, we disregard the ice sensible heat uptake as small relative to the latent heat uptake. We make a case that in analysing the causes of model differences in ice melt/growth, it is sufficient in most parts of the Arctic to analyse the causes of differences in the net gridbox mean surface flux (over ice- and ocean-covered portions of grid cells). That is, the gridbox mean net surface flux can be treated as the primary driver, to first order, of the ice melt and growth.

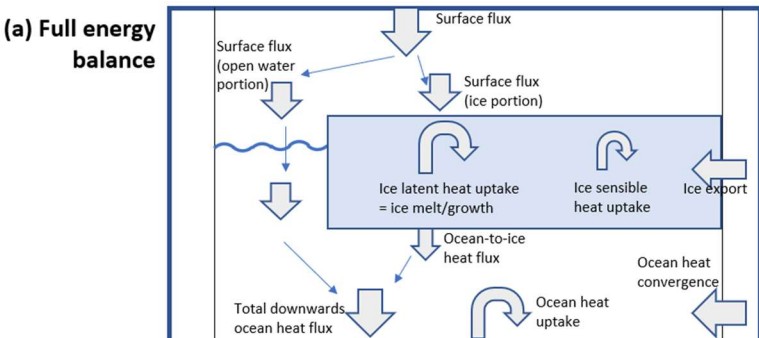

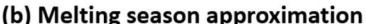

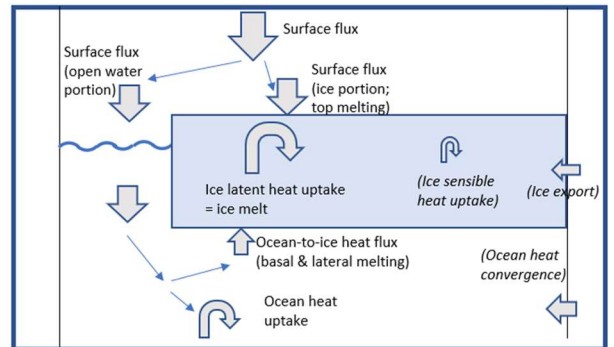
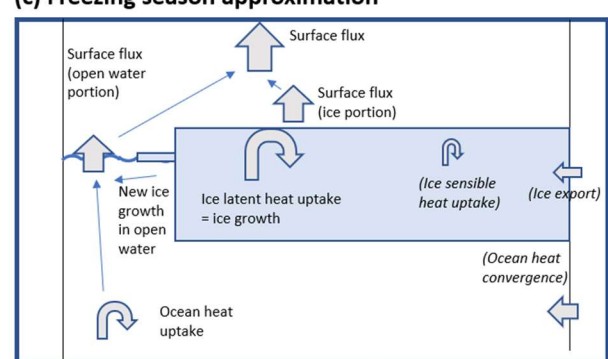

**Figure 2. Schematic showing (a) the principal energy flows between atmosphere, ice and ocean in the Arctic; (b) how surface flux relates to ice melt/growth in the melting season; (c) as (b), for the freezing season.**

Firstly examining the melting season (Figure 2b), the melting of the snow-ice column from the top is directly analogous to the portion of the surface flux occurring over ice-covered regions. Basal and lateral melting, by contrast, are driven by ocean-to-ice heat flux: evidence from models (Steele et al., 2010; Keen and Blockley, 2018) and observations (McPhee et al., 2003; Perovich et al., 2008) suggests that this derives almost entirely from direct solar heating rather than from oceanic heat convergence. Hence in our conceptual picture, the portion of surface flux over open sea is converted into both basal and lateral sea ice melt, and into sensible heating of the top ocean layer.

In the freezing season (Figure 2c), the growth of the snow-ice column is primarily driven by cooling from the top, analogous to the portion of the surface flux occurring over ice-covered regions. The open water part of the surface flux is associated firstly with cooling of the top ocean layer, and secondly with the formation of new ice.

In summary, over the year as a whole the net surface flux is analogous to the melting/freezing of the sea ice, together with warming/cooling of the top ocean layer. Hence the net surface flux can be treated as the primary driver of the sea ice melt/growth.

To further inform this picture, we calculate Arctic Ocean averaged ice export and oceanic heat convergence for the period 1980-1999 in the evaluated models. The ice divergence term is small, with indeterminate annual cycles, at 2.5, 4.0 and 5.1 $Wm^{-2}$ in HadGEM2-ES, HadGEM3-GC3.1-LL and UKESM1.0-LL respectively. The oceanic heat convergence term is also small at 4.4 $Wm^{-2}$, 3.8 $Wm^{-2}$ and 3.9 $Wm^{-2}$ respectively; in all models, these values show high sensitivity to the location of the Arctic Ocean region boundary in the Atlantic sector, suggesting that most of this heat is released close to the Atlantic ice edge. Monthly differences in these terms between models are too small to explain the differences in ice melt / growth seen: with ice density of $917 kgm^{-3}$ and latent heat of fusion of $3.35 \times 10^3$ $Jkg^{-1}$, a difference of $1 Wm^{-2}$ is equivalent to only an additional 8.4mm of ice melt per month. This supports the view that the surface flux should be treated as the primary driver of ice melt/growth, and that causes of differences in ice melt/growth should be sought first by analysing this term. Hence our next step is to evaluate and compare surface radiative fluxes, the largest components of the surface flux.

### 3.3 Surface radiation

We now evaluate surface radiative fluxes in the three models (Figure 3). Downwelling SW is very similar in all three models year-round (Figure 3a). By contrast upwelling SW displays large differences from June-August (Figure 3b), with HadGEM2-ES lowest in magnitude and UKESM1.0-LL highest. This is consistent with the ice melt/growth differences: lower upwelling SW in HadGEM2-ES causes a higher net SW flux (Figure 3c), and more overall sea ice melting. It is also consistent with the ice extent differences: the lower summer ice extent of HadGEM2-ES causes a lower surface albedo and hence lower SW reflection. However, in June the ice area differences are small, and insufficient to explain the upwelling SW

differences. Hence another parameter influencing surface albedo is likely to be responsible for the model differences in this month. We examine possible drivers shortly.

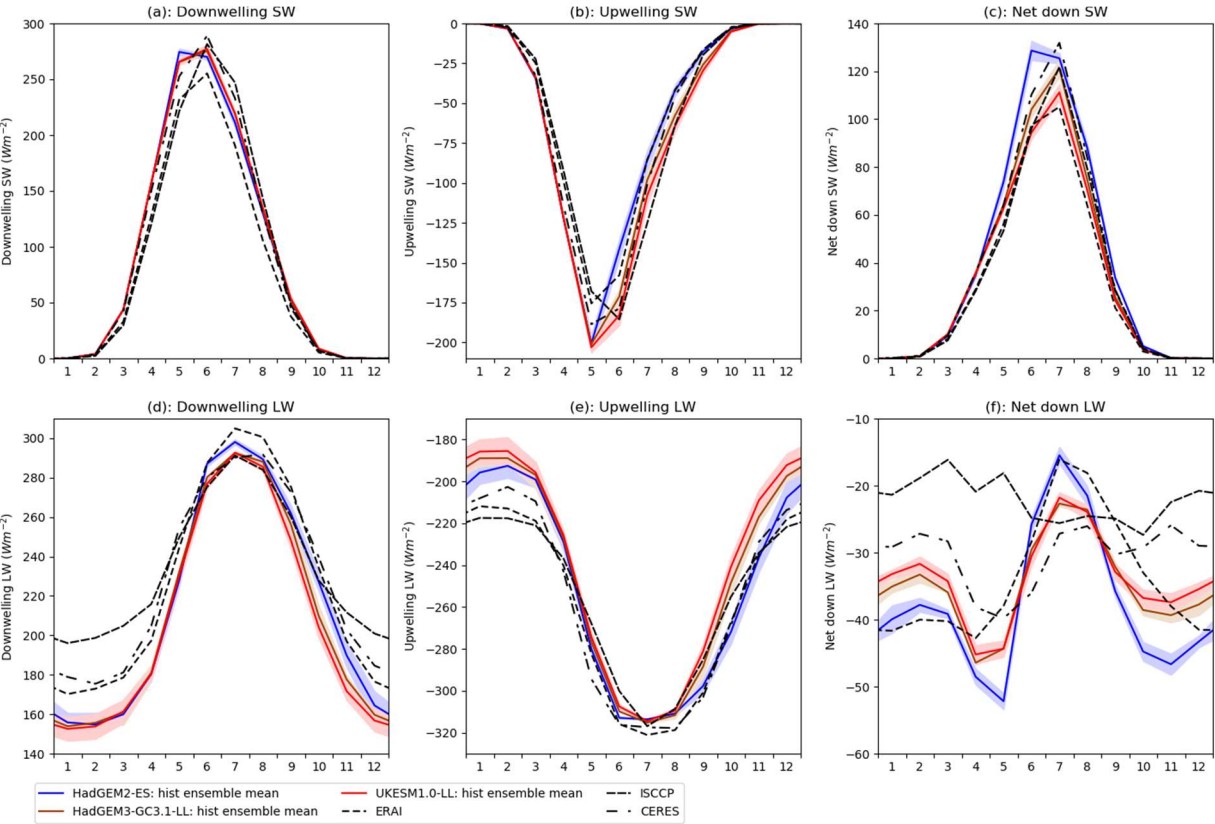

**Figure 3. (a) Downwelling SW, (b) upwelling SW, (c) net downwards SW, (d) downwelling LW, (e) upwelling LW, (f) net downwards LW, from 1980-1999 in HadGEM2-ES, HadGEM3-GC3.1-LL and UKESM1.0-LL, together with observational estimates from ISCCP-FD, ERAI and CERES, averaged over the Arctic Ocean region. For each model, ensemble mean (lines) and twice standard deviation (shading) is shown.**

Downwelling LW fluxes are similar from January-May in the three models (Figure 3d), but from June-July, and October-November downwelling LW is much higher in HadGEM2-ES than in UKESM1.0-LL and HadGEM3-GC3.1-LL, indicating a colder, or clearer, atmosphere in these months in the newer models. Upwelling LW fluxes are higher in magnitude in HadGEM2-ES than in UKESM1.0-LL and HadGEM3-GC3.1-LL from September – February (Figure 3e), indicating a

colder surface in the newer models. Net LW is higher in HadGEM2-ES from June-August, and lower from September – May, than in the newer models (Figure 3f), indicating that the upwelling LW differences dominate during the winter. The net LW differences are also consistent with the greater ice melting and ice freezing seen in HadGEM2-ES relative to the CMIP6 models. For both net LW during winter, and net SW during summer, model internal variability is small relative to the difference between HadGEM2-ES and the CMIP6 models.

To summarise, the weaker summer ice melt of the CMIP6 models relative to HadGEM2-ES is driven by a smaller upwelling SW flux from June – August. This is likely caused by ice area differences in July and August (the surface albedo feedback), but in June other surface albedo drivers are responsible. The weaker winter ice growth of the CMIP6 models relative to HadGEM2-ES is driven by a smaller upwelling LW flux from October – April. This is likely caused by the ice thickness-growth feedback: compared to HadGEM2-ES, the two CMIP6 models have thicker ice which leads to a colder surface due to reduced heat conduction through the ice, and the colder surface results in less longwave radiative loss to space.

### 3.4 Variables influencing surface albedo

We now evaluate other model variables that affect surface albedo. Surface albedo $\alpha_{sfc}$ can be expressed as

$$\alpha_{sfc} = \sum_i A_i \alpha_i \qquad\qquad (1)$$

where $i$ varies over all surface types present in a grid cell, $A_i$ is the fractional area of surface type $i$ and $\alpha_i$ is the surface albedo of surface type $i$. Hence differences in surface albedo between models must be associated either with differences in area of surface types, or differences in their albedo. For all evaluated models, the relevant surface types are open water, snow, bare ice or meltpond. The ice fraction (which contains the snow, bare ice and meltpond surface types) in all models evolves as a prognostic variable, but within the sea ice the snow and meltpond fractions are subject to different parameterisations, discussed in more detail below. In seeking the cause of the surface albedo differences, we concentrate primarily on differences in snow and meltpond fraction. Furthermore, we concentrate on the month of June as it is in this month that the models display differences in upwelling SW radiation that cannot be explained by ice area differences alone.

Firstly we compare the sea ice area fraction in the three models (Figure 4a-c). Ice area is very similar (close to 1) away from the Arctic Ocean coasts in all three models; only near the Fram Strait and the Barents Sea does HadGEM2-ES display significantly lower values. This is consistent with the ice area differences being insufficient to explain the upwelling SW differences. For comparison, we show HadISST1.2 sea ice area observations from 1980-1999 (Figure 3d), which are consistent with the three models across much of the Arctic, and near the Fram Strait and Barents Sea display values higher than HadGEM2-ES but lower than the CMIP6 models.

Secondly, we compare snow cover in the three models. In June, snow thickness is highest in UKESM1.0-LL and lowest in
HadGEM2-ES (Figure 4e-g), with Arctic Ocean average snow thicknesses of 6cm, 13cm and 18cm respectively. For each
model, ensemble standard deviation snow thickness is mostly lower than 2cm, although in UKESM1.0-LL standard
deviation approaches 5cm between Fram Strait and the North Pole (not shown).

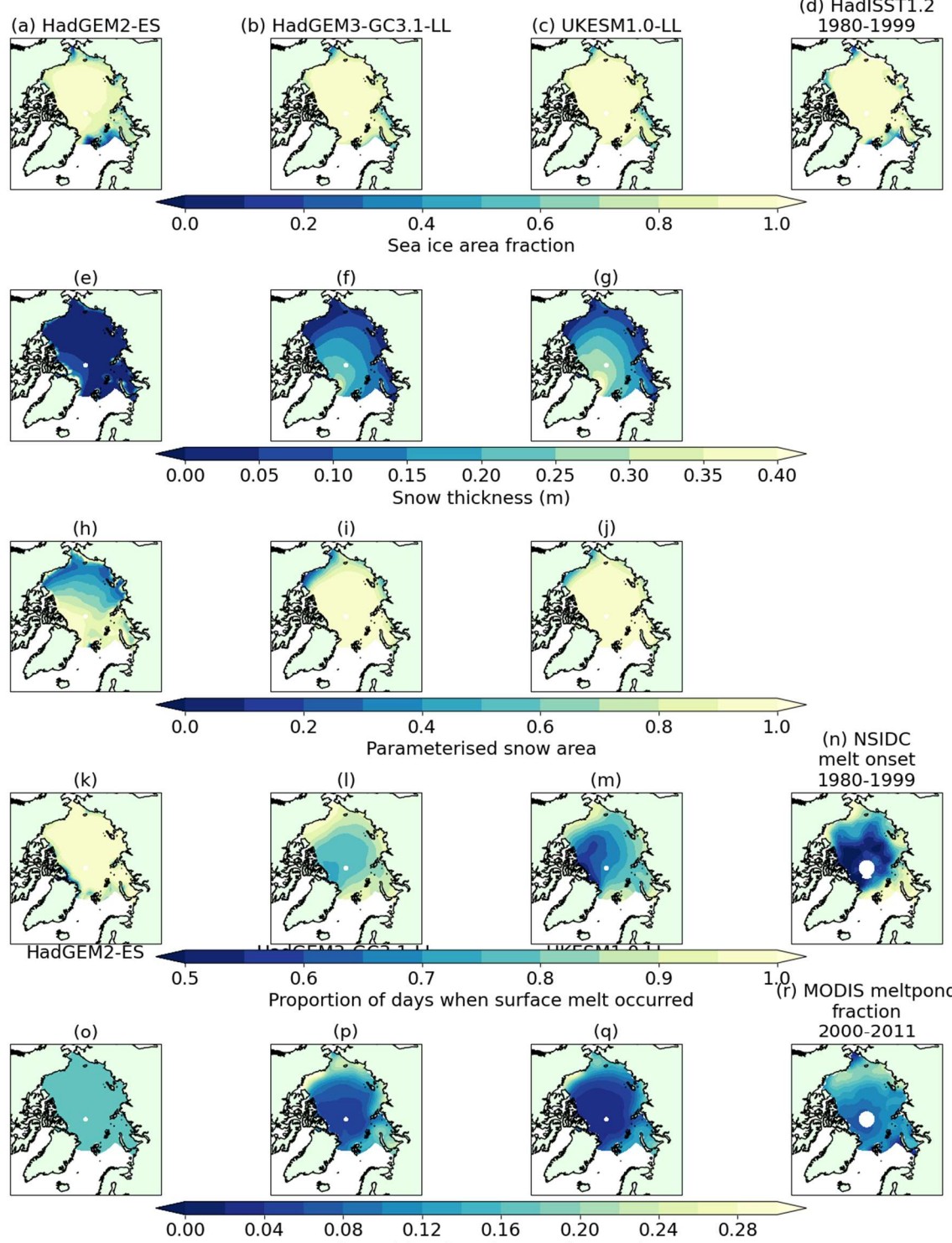

(a) HadGEM2-ES    (b) HadGEM3-GC3.1-LL    (c) UKESM1.0-LL    (d) HadISST1.2
1980-1999

Sea ice area fraction
0.0    0.2    0.4    0.6    0.8    1.0

(e)    (f)    (g)

Snow thickness (m)
0.00    0.05    0.10    0.15    0.20    0.25    0.30    0.35    0.40

(h)    (i)    (j)

Parameterised snow area
0.0    0.2    0.4    0.6    0.8    1.0

(n) NSIDC
melt onset
1980-1999

(k)    (l)    (m)

HadGEM2-ES    HadGEM3-GC3.1-LL    UKESM1.0-LL

Proportion of days when surface melt occurred
0.5    0.6    0.7    0.8    0.9    1.0

(r) MODIS meltpond
fraction
2000-2011

(o)    (p)    (q)

Meltpond fraction given surface melt
0.00    0.04    0.08    0.12    0.16    0.20    0.24    0.28

**Figure 4. 1980-1999 June averages of variables affecting surface albedo in HadGEM2-ES (first column), HadGEM3-GC3.1-LL (second column) and UKESM1.0-LL (third column), compared where appropriate to observational datasets (fourth column). Showing (a-d) sea ice area fraction, (e-g) snow thickness, (h-j) parameterized snow area fraction, (k-n) fraction of month during which surface melting conditions were present, (o-r) melt-pond area fraction during times in which surface melting conditions were present.**

The models differ in how snow area is parameterized from snow thickness. HadGEM2-ES uses the formula

$$A_{snow} = 1 - e^{-0.2 h_{snow} \rho_{snow}} \tag{2}$$

while HadGEM3-GC3.1-LL and UKESM1.0-LL use the formula (from CICE)

$$A_{snow} = \frac{h_{snow}}{h_{snow} - 0.02} \tag{3}$$

where $A_{snow}$, $h_{snow}$ and $\rho_{snow}$ refer to snow area, thickness and density respectively.

The effect of this is that the newer models simulate a lower fraction of snow for the same snow thickness value, with the difference greatest at a thickness of 4cm when the newer models simulate 26% less snow area (Figure S1). We can see the effect of this when we compare maps of June snow thickness in the three models (Figure 4e-g) with maps of June snow area estimated using the models' respective formulations (Figure 4h-j). Despite the substantially higher snow thicknesses in HadGEM3-GC3.1-LL and UKESM1.0-LL, the increase in snow area in the newer models is muted. The effect of the new parameterization is to reduce the surface albedo difference caused by the thicker snow in UKESM1.0-LL and HadGEM3-GC3.1-LL Despite this, the difference in snow area between HadGEM2-ES and the CMIP6 models (rising to 0.7 near the Bering Strait) is much larger than the ensemble standard deviations in snow area, which approach 0.15 near the Bering Strait in HadGEM2-ES but are otherwise under 0.05 (not shown).

Next, we examine differences in melt-pond coverage between the models. As with the snow comparison, care is required as the explicit topographic melt-pond scheme of the CMIP6 models is more complex than the simple parameterization of HadGEM2-ES, in which (broadband) albedo is linearly reduced from 0.80 to 0.65 (snow) or from 0.61 to 0.52 (bare ice) as surface temperature rises from -2°C to 0°C. We first compare the average date of melt onset between the three models: the first day, for each grid cell and year, for which the surface temperature exceeds a fixed threshold, chosen to be -1C. In this way we create a metric to compare surface melting tendencies in each model, independent of the melt pond scheme.

Date of melt onset is earliest in HadGEM2-ES, at mid-to-late May in the Central Arctic, and latest in UKESM1.0-LL, in late June in the Central Arctic. By comparison, SSMI observations suggest melt onset occurred in mid-June, on average, in the

Central Arctic for the period 1980-1999. Hence UKESM1.0-LL models melt onset to occur too late in the Central Arctic, but HadGEM2-ES is much too early (as identified by West et al., 2019). To demonstrate the effect of this on June surface albedo, the proportion of June for which melting conditions are present ($T_{melt}$) is plotted (Figure 4k-m): for HadGEM2-ES, $A_{melt}$ is 100% across most of the Arctic, but in HadGEM3-GC3.1-LL and UKESM1.0-LL $T_{melt}$ approaches 70% and 50% respectively across the central Arctic. Ensemble standard deviation in $T_{melt}$ is nowhere higher than 10% for any model (not shown). For reference, we show average proportion of time in which surface melting was present derived from the NSIDC satellite dataset described in Section 2 (Figure 3n). This demonstrates that HadGEM2-ES and HadGEM3-GC3.1-LL tend to model too much surface melting in June, while UKESM1.0-LL is much closer to observations.

To assess the effect of melt-ponds on surface albedo, the melt-pond fraction where melting is taking place must also be compared. In this way the proximate effect of the different melt-pond schemes is captured. HadGEM2-ES effectively simulates a uniform, constant melt-pond fraction of 0.2 in the presence of surface melting, while UKESM1.0-LL and HadGEM3-GC3.1-LL simulate variable melt-pond fraction depending on total meltwater and ice topography. Hence June melt-pond fractions are higher in the CMIP6 models than HadGEM2-ES at lower latitudes, but much lower in the Central Arctic (Figure 4o-q). Ensemble standard deviation in the CMIP6 models is everywhere less than 4% (not shown). Hence the explicit melt-pond scheme is contributing to a portion of the surface albedo differences in June, in addition to the different melt onset dates and snow thickness. For reference, we also show MODIS-derived meltpond fraction for 2001-2011 (Figure 3r; data is unfortunately not available for the reference period 1980-1999). This demonstrates that the uniform meltpond fraction effectively assumed by HadGEM2-ES is likely to be too low near the Arctic Ocean coasts and too high in the interior. While the modelled meltpond fraction in the CMIP6 models is generally lower than MODIS, this is almost certainly partly due to MODIS being from a later period, in which surface melting of Arctic sea ice began earlier in the year (e.g. Markus et al., 2011).

In summary, differences in both snow and meltpond cover during June are consistent with surface albedo differences between the three models. We now quantify the approximate surface albedo difference between UKESM1.0-LL and HadGEM2-ES that we expect due to the model differences in ice area, snow thickness, snow area parameterisation, surface melting tendency, and meltpond parameterisation. For the ice area, snow thickness and surface melting tendency we do this by multiplying the model difference in each variable in question by the change in model mean surface albedo that would be associated with that variable. For the parameterisation variables we do this by calculating surface albedo in the model mean state according to both model parameterisations, and taking the difference, This gives five surface albedo difference 'components' whose sum is very similar in magnitude and spatial pattern to the actual June surface albedo difference (Figure 5). Surface albedo is everywhere higher in UKESM1.0-LL than in HadGEM2-ES (except on the Beaufort and Kara Sea coasts), but the component analysis demonstrates that this is due to different causes in different parts of the Arctic. On the ice margins, and particularly in the Atlantic sector, the ice area difference dominates; in the coastal Arctic Ocean seas, the snow thickness difference dominates; in the central Arctic, surface melting tendency and meltpond parameterisation

dominate. The effects combine to make surface albedo much higher everywhere in UKESM1.0-LL than in HadGEM2-ES, despite the snow area parameterisation difference creating an opposing effect.

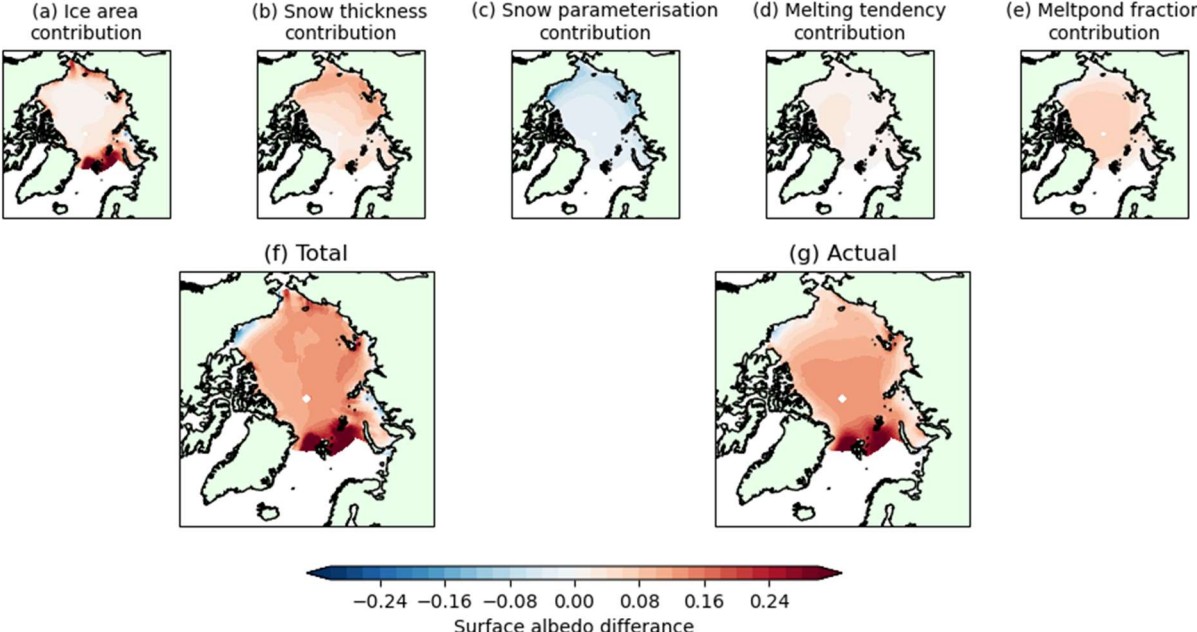

**Figure 5. Components of June surface albedo difference for UKESM1.0-LL relative to HadGEM2-ES (1980-1999, averaged over all ensemble members). Shown are estimated surface albedo difference due to (a) ice area difference; (b) snow thickness difference; (c) the different snow area parameterisations used by the two models; (d) difference in surface melting tendency, or proportion of time during which surface melting conditions were present; (e) the different meltpond parameterisations used by the two models. (f) shows the total surface albedo difference produced by all components, and (g) shows the actual surface albedo difference between the two models, diagnosed from downwelling and upwelling SW radiation, for comparison.**

The component analysis is a powerful way to examine proximate causes of differences in surface albedo, and hence in surface flux. However, a shortcoming is that not all surface albedo differences shown here are equal. An albedo difference of 0.15 will have a greater effect on surface flux, and hence on ice melt/growth, if it occurs in a region of greater downwelling SW radiation. In other words, the surface albedo differences do not operate linearly on the ice melt/growth, and cannot be averaged over space and time to estimate meaningful surface flux or ice melt/growth differences. This observation motivates the ensuing analysis.

**4 Attributing differences in modelled ice melt/growth using an induced surface flux (ISF) difference framework**

The model evaluation presented in Section 3 prompts two questions. Firstly, by how much do the model differences in snow

and meltpond variables affect the ice melt differences over the melting season? Secondly, does the thickness-growth feedback fully account for the differences in ice growth, given the opposing effect of the downwelling LW differences during the early winter? To address these questions we use a method similar to the 'induced surface flux' (ISF) framework described by West et al. (2019). This framework attempts to separate and quantify the drivers of differences in surface flux, and hence in seasonal ice growth/melt, of which the surface flux is treated as the principal driver. In a similar way to the

surface albedo analysis above, it aims to split model differences in surface flux into components attributable to specific model differences. Crucially, when surface flux rather than surface albedo is treated as the dependent variable, components can be averaged in time and space to obtain meaningful aggregate statistics, comparable to ice melt/growth differences.

The ISF analysis works by approximating, at any point $(\boldsymbol{x}, t)$ of model space and time, the total net surface flux as a function $g(v_i, \boldsymbol{x}, t)$ of variables $v_i$ which are 'quasi-independent'. We loosely define quasi-independence to mean that the variables

affect surface flux on a much shorter timescale than that in which the affect each other. (For example, two such variables could be downwelling SW radiation and ice concentration, as in Holland and Landrum, 2015). In this way the rate at which surface flux depends upon a model variable $v_i$ at point $(\boldsymbol{x}, t)$ can be estimated as $\partial g / \partial v_i$. Given two model estimates of $v_i$ at point $(\boldsymbol{x}, t)$, $v_i^{MODEL1}(\boldsymbol{x}, t)$ and $v_i^{MODEL2}(\boldsymbol{x}, t)$, we can characterise the induced surface flux difference between the models due to variable $v_i$ at $(\boldsymbol{x}, t)$ as

$$\left(v_{i,x,t}^{MODEL1} - v_{i,x,t}^{MOD}\right)\left(\frac{\partial g}{\partial v_i}^{MODEL1} + \frac{\partial g}{\partial v_i}^{MODEL2}\right)/2 \quad (4)$$

The ISF differences obtained can be averaged over large regions of time and space to determine the large-scale effects of specific model biases on surface flux. In particular, the quasi-independence requirement helps refine causality. Due to the requirement for the variables to affect surface flux on shorter timescales than they affect each other,, the ISF differences for each variable should represent largely separate effects, and the sum of the individual ISF differences should approach the

355 true surface flux difference, which is thereby divided into components, each representing the effect of a model variable. In turn, this allows the drivers of differences in ice melt/growth to be separated and quantified.

We briefly describe how the effects of different model variables on surface flux are quantified in this way (West et al., 2019 gives a more complete description). Firstly, the division of the upwelling SW difference into components is demonstrated. Net downwelling SW radiation can be expressed as

$$F_{SW-net} = F_{SW-down} - F_{SW-up}$$

$$= F_{SW-down}\left(1 - \alpha_{sfc}\right) \qquad (5)$$

where $\alpha_{sfc}$ is surface albedo, $F_{SW-do}$ is downwelling SW radiation and $F_{SW-up}$ is upwelling SW radiation. $\alpha_{sfc}$ can be further expressed as

$$\alpha_{sfc} = A_{ocean}\alpha_{ocean} + A_{bare\_ice}\alpha_{bare\_ice} + A_{snow}\alpha_{snow} + A_{pond}\alpha_{pond} \tag{6}$$

where $A_i$ is the fractional area of surface type $i$ and $\alpha_i$ is the surface albedo of surface type $i$. As $\sum_i A_i = 1$, the $A_i$ are not quasi-independent variables (a change in one entails a change in one or more of the others), hence we substitute $A_{bare-ice} = A_{ice} - A_{snow} - A_{pond}$ and $A_{ocean} = 1 - A_{ice}$, where $A_{ice}$ is the total area covered by sea ice in the grid cell. This then expresses $\alpha_{sfc}$ in terms of $A_{ice}$, $A_{snow}$ and $A_{pond}$, allowing the impact of ice, snow and meltpond differences on surface flux to be examined separately.

We refine the calculation further in two ways. Firstly, to separate the melting tendency $T_{melt}$ (as defined in Section 3.3) from the effects of the different meltpond parameterisations, we define $A_{meltpond-given-melti} = A_{meltpond}/T_{melt}$, and substitute $A_{meltpond} = T_{melt}A_{meltpond-given-meltin}$ into equation (6). Because $T_{melt}$ represents the response of the surface temperature to atmospheric forcing, and $A_{meltpond-given-melting}$ the response of the meltpond scheme to melting conditions, these variables also represent quasi-independent, separate effects on the surface flux. Specifically, in HadGEM2-ES the meltpond parameterisation effectively assumes uniform, constant $A_{meltpond-given-melting}$ of 0.23 (when snow is present) or 0.18 (when snow is not present), while in the CMIP6 models this quantity can vary.

Secondly, to separate the effect of model differences in snow thickness (the actual prognostic snow variable) from that of the different parameterisations of snow area, we write $A_{snow} = f_{MODEL}(h_{snow})$, where $f_{MODEL-i}$ are Equations (2) for HadGEM2-ES and (3) for UKESM1.0-LL and HadGEM3-GC3.1-LL.

In summary,

$$F_{SW-net} = F_{SW-dow}\left(1 - \alpha_{ocean}(1 - A_{ice}) - \alpha_{snow}f_{MODEL}(h_{snow}) - \alpha_{meltpond}T_{melt}A_{meltpond-given-melting} - \right.$$
$$\left. \alpha_{bare-ice}\left(A_{ice} - f_{MODEL-i}(h_{snow}) - T_{melt}A_{meltpond-given-melting}\right)\right) \tag{7}$$

In this way, the surface flux is expressed as a function of downwelling SW, ice area, snow thickness, snow parameterisation melting 'tendency' and meltpond fraction. These variables have the required property of affecting surface flux on a shorter timescale than that which on they affect each other. Indeed, their effect on surface flux via the net SW flux (mainly via surface albedo) is near-instantaneous.

We now describe how albedo parameters are prescribed. Open sea albedo is parameterised in HadGEM2-ES according to Barker and Li (1995) and in the CMIP6 models according to Jin et al. (2011); instead of attempting to replicate this in our simple model, we calculate the distribution of open sea albedo values from model SW radiation diagnostics, and prescribe ocean albedo in our simple model to be 0.07, close to the modal value seen. For the CMIP6 models, meltpond, bare ice and

snow albedos are prescribed as separate infrared and visible values, respectively 0.07 and 0.27 for meltponds, 0.36 and 0.78 for bare ice and 0.78 and 0.98 for snow; in our simple model, we combine these values using an infrared fraction of 0.4. In HadGEM2-ES, bare ice and snow albedos are 0.61 and 0.80 respectively.

For any model, month and grid cell, in addition to approximating the net SW flux, we can then characterise the dependence of the net SW flux on any model variable or parameter by taking the derivative of equation (7) with respect to that parameter. For example, net SW varies with ice concentration by $F_{SW-down}(\alpha_{ocean} - \alpha_{bare-ice})$. As a second example, the dependence of net SW on $T_{melt}$, the tendency of a grid point to be undergoing surface melting, is

$$\frac{\partial F_{SW-ne}}{\partial T_{melt}} = F_{SW-down} A_{meltpond-given-melti} \quad \left(\alpha_{meltpond} - \alpha_{bare-ice}\right) \quad (8)$$

The contribution of the snow parameterisation scheme is calculated differently to that of the other components, by evaluating surface flux at the model mean state using each snow area function in turn, and taking the difference.

In general therefore, given a model difference in a variable at any point in model space and time, we multiply that difference by the model mean dependence of net SW on that variable, to obtain an estimate of the surface flux difference induced by the difference in that variable. In this way, the model differences in upwelling SW can be decomposed into contributions from different model parameters, and the contributions can be averaged in time and space to determine the large-scale effects of differences in each variable.

We demonstrate this by showing how components of induced surface flux difference evolve through the summer when comparing the models in turn. As daily data is available for only the first ensemble member of HadGEM2-ES, the evolution is demonstrated using only first historical ensemble members when comparing to this model. As the ensemble spread in the driving model variables is small relative to the model differences (demonstrated in Section 3.3), this is likely to give a similar result to using the ensemble means.

When comparing UKESM1.0-LL with HadGEM2-ES (Figure 6a), the total ISF difference is negative throughout the summer, consistent with the weaker ice melt and reduced net SW radiation in UKESM1.0-LL. In April, the total ISF difference is weakly negative, but it falls steeply through May to reach values of over -30 Wm$^{-2}$ in mid-June, rising very slowly thereafter. Consistent with the evaluation in Section 3, ice area difference does not become the dominant contribution to the total ISF difference until early July, with significant contributions occurring both from surface melt onset occurrence (peaking at the beginning of June, at -15 Wm$^{-2}$) and from snow area (peaking in mid-to-late June, at -18 Wm$^{-2}$). The melt-pond scheme contributes only a small negative ISF difference, because large negative differences in the Central Arctic are mostly outweighed by large positive differences near the coasts. The difference in snow parameterization, as expected, contributes a positive ISF difference, rising to a maximum of 8 Wm$^{-2}$ in early June. The downwelling SW term contributes a negative difference in May, becoming a positive difference in June, but this is counteracted by the downwelling LW term

which is shown for comparison. These terms likely represent the response of modelled cloud cover to the weaker ice melt of UKESM1.0-LL, and collectively contribute little surface flux difference.

Examining the ISF terms over the melt season as a whole (Figure 6b) we see that the ice area term contributes about two-thirds of the total ISF difference (18.6 Wm-2 of 27.2 Wm-2), with the snow area and melt onset terms contributing -3.2 Wm-
2 and -2.4 Wm-2 respectively. The value of using daily data is demonstrated: the snow area and melt onset terms appear very small over the season as a whole, but the daily data shows that they play a vital role in the early part of the season, and help generate the ice area differences that contribute the major part of the surface flux differences. The downwelling radiative differences collectively contribute -4.9 Wm-2, but the effect is much more uniform over the melting season than that of the snow area and melt onset terms.

Spatial maps of the ISF components show a very similar picture to the surface albedo 'decomposition' in Figure 4. In June for example, the total ISF difference is everywhere strongly negative (indicating less melt in UKESM1.0-LL), by 20-40 Wm$^-$$^2$, but this arises as the sum of terms which dominate in different regions: the ice area term at the coasts, the snow thickness term further into the Arctic Ocean and the meltpond terms in the Central Arctic. The region in which the snow thickness ISF component dominates, however, extends further into the Central Arctic than in the surface albedo comparison. This may be a
result of using higher-resolution, daily data for the ISF calculation. The downwelling SW and downwelling LW components display nearly equal and opposite patterns in all months, except July when there is a substantial SW+LW radiative difference of -10 Wm$^{-2}$ in a large region of the Central Arctic that is also likely implicated in the weaker sea ice melt of UKESM1.0-LL.

When HadGEM3-GC3.1-LL is compared to HadGEM2-ES (not shown) the picture is qualitatively similar but the ISF terms
are smaller in magnitude, consistent with the rate of ice melt being more similar in these two models. The total ISF difference is smaller still when UKESM1.0-LL is compared to HadGEM3-GC3.1-LL (Figure 6c), but in this case the snow thickness component is comparable in magnitude to the ice area component(Figure 6d), . This occurs because despite the snow thickness component being of similar size in the two comparisons, the ice area component is much larger when UKESM1.0-LL is compared to HadGEM2-ES than when compared to HadGEM3-GC3.1-LL. This likely reflects the
nonlinear relationship between ice thickness and ice area: more ice area is lost during a melting season at lower ice thicknesses. Hence the ice area term naturally plays a proportionally larger role in differences wrt HadGEM2-ES (with the thinnest ice) than wrt HadGEM3-GC3.1.

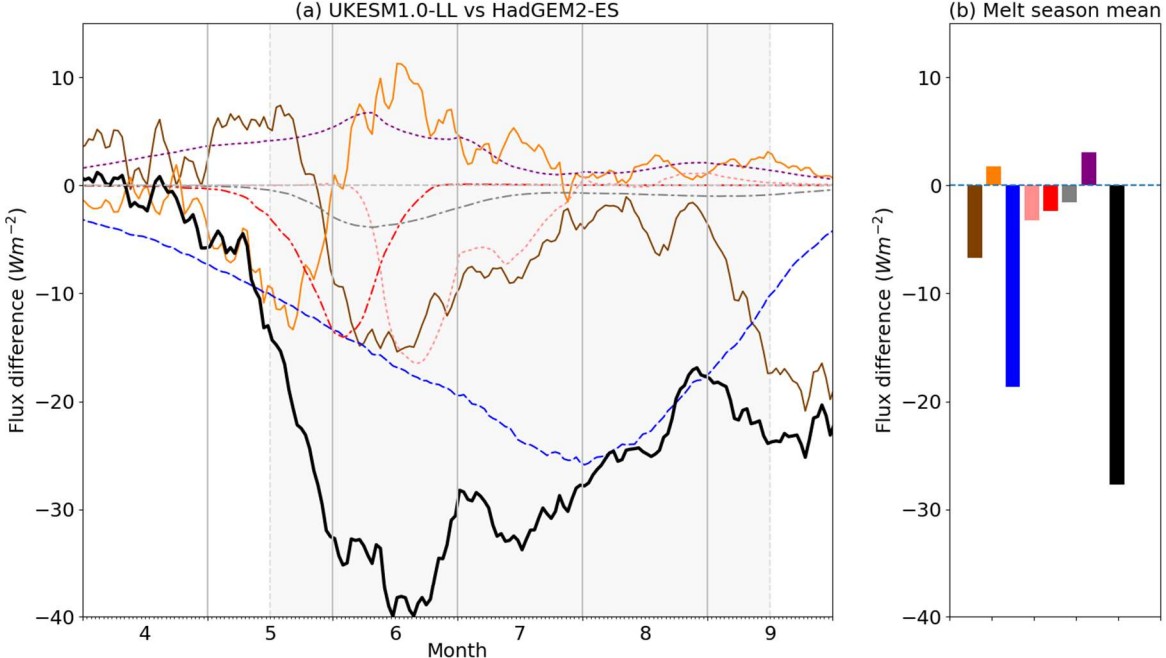

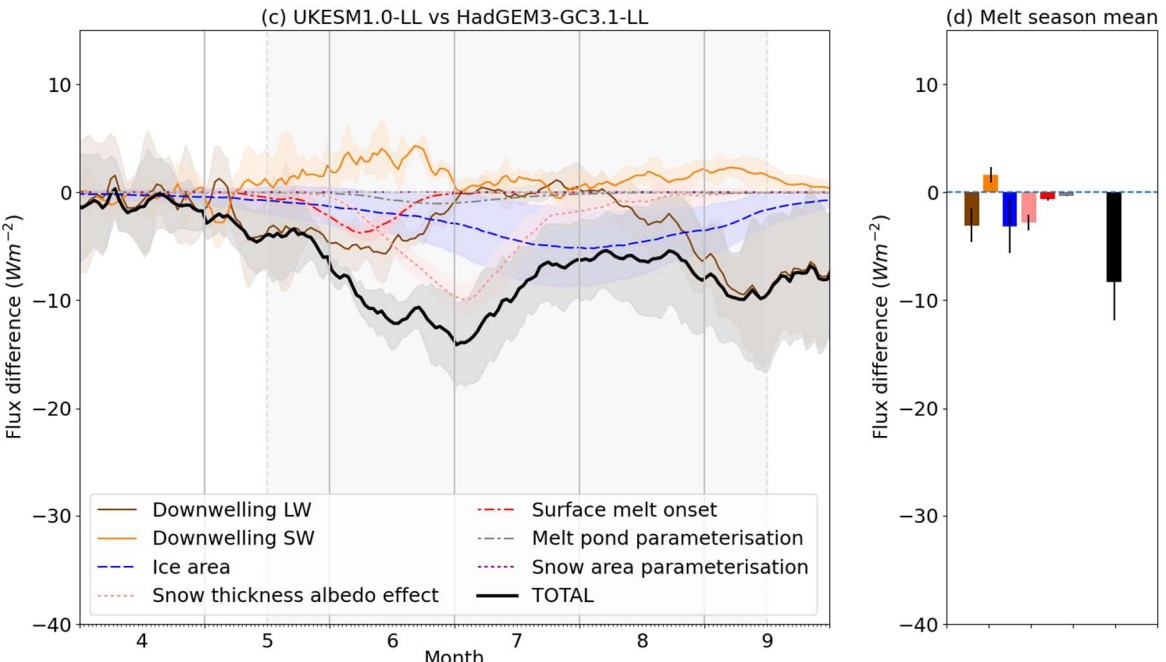

**Figure 6. Illustration of drivers of summer melting differences, using the ISF framework. Shown are ISF differences between (a,b) UKESM1.0-LL and HadGEM2-ES; (c,d)  UKESM1.0-LL and HadGEM3-GC3.1-LL, apportioned by downwelling SW, ice area, snow thickness, surface melt onset, direct impact of snow fraction parameterization, and direct impact of the UKESM1.0 melt-pond scheme. Showing April – September daily evolution (left column) and melt season mean (right column, 16th May – 15th September). For the first comparison, only the first ensemble member is shown, as daily data is not available for other members of HadGEM2-ES; for the second comparison, all four ensemble members of both models are used, with ensemble mean and standard deviation indicated.**

Secondly, we describe how the thickness-growth feedback can be separated from the atmospheric radiative components. This is done using a single-column model after Thorndike (1992), discussed more extensively in West et al. (2019). As in the models, each grid cell is divided into five sea ice categories, plus a 'zero' category for open water. For each category, we assume a uniform conductive flux through the ice (no sensible heat storage) and flux continuity at the surface. Hence

$$F_{cond}(x, t, cat) = \frac{T_{sfc}(x,t,cat) - T_{base}}{R_{ice}(x,t,cat)} \quad (9)$$

where $cat$ represents the category number (ranging from 0 to 5), $F_{cond}$ is conductive flux through the ice, $T_{base}$ is ice base temperature, $T_{sfc}$ ice surface temperature. $R_{ice}$ denotes the thermal insulance of the ice-snow column, defined as

$$R_{ice} = \frac{h_{ice}(x,t,cat)}{k_{ice}} + \frac{h_{snow}(x,t,cat)}{k_{snow}} \quad (10)$$

where $h_{ice}$ and $h_{snow}$ are category ice and snow thickness respectively, and $k_{ice}$ and $k_{snow}$ are fresh ice and snow conductivity. The latter are parameterized as 2.03 and 0.30 Wm$^{-1}$K$^{-1}$ respectively, similar to the values used in HadGEM2-ES, HadGEM3-GC3.1-LL and UKESM1.0-LL.

By flux continuity, $F_{cond} = F_{sfc}$, where $F_{sfc}$ is the downwards surface energy flux. We linearise the dependence of $F_{sfc}$ on surface temperature $T_{sfc}$ as $F_{sfc}(x, t, cat) = F_{atmos-ice}(x, t) + B_{up}T_{sfc}(x, t, cat)$. For optimal estimation, the linearisation is at each grid cell centred at $T_{sfc-0}(x, t)$, the monthly mean surface temperature at that grid cell averaged between the two models being compared.

Eliminating $T_{surface}$ and rearranging gives an equation for $F_{sfc-c}$ , the surface flux over a single ice category:

$$F_{sfc-c} = \frac{F_{atmos-ic} + B_{up}T_{base}}{1 + B_{up}R_{ice}} \quad (11)$$

Summing over categories gives

$$F_{sfc-ice} = F_{atmos-ice} \sum_{cat=1}^{5} \frac{a_{ice}}{1 + B_{up}R_{ice}} \quad (12)$$

where $F_{sfc-i}$ represents the total surface flux over the ice-covered portion of the grid cell. In this equation $F_{atmos-ic}$ represents the part of the atmosphere-ice energy flux that is independent of the surface temperature, or the components that do not vary on short timescales when surface temperature is varied.

After West et al. (2019) we identify $F_{atmos-ice}$ with the sum of the net SW flux, the downwelling LW flux, and the turbulent fluxes over the ice-covered portion of the grid cell. The net SW flux is estimated as in equation (5), while the downwelling

LW flux (invariant over the grid cell) and the turbulent fluxes over ice are obtained from model diagnostics.

We justify treating the turbulent fluxes as independent of the surface temperature because surface temperature adjustments tend to occur in association with similar near-surface air temperature adjustments, and therefore cause little turbulent flux variation on short timescales. We justify treating downwelling LW as independent of the surface temperature because the processes by which surface temperature changes influence downwelling LW are too complex to include in our model,

involve likely discrete changes in boundary layer cloud properties (e.g. Morrison et al., 2012), and occur over timescales at least as long as those over which airmasses are replaced and are therefore less likely to be relevant.

To obtain the total surface flux we add the surface flux over the open water portion of the grid cell to obtain

$$F_{sfc} = (1 - \sum_{cat=1}^{5} a_{ice})F_{atmos-ocean} + F_{atmos-ic} \sum_{cat=1}^{5} \frac{a_{ice}}{1+B_{up}R_{ice}} \quad (13)$$

Here $F_{atmos-ocean}$ represents the total surface flux over the non-ice covered portion of the grid cell, and is calculated

similarly to $F_{sfc-i}$ but using the open water values of net SW and turbulent fluxes.

In this way, we approximate surface flux as a function of downwelling SW, downwelling LW, sensible and latent heat fluxes, and of category ice area, ice thickness and snow thickness. In HadGEM2-ES each of these affects surface flux instantaneously in a manner analogous to our simple model. Although in the CMIP6 models the ice heat capacity complicates the surface heat flux response, it is still the case that for each variable the effect on the surface heat flux must be

realised before the effect on another variable. For example, over longer timescales changes in ice thickness would affect downwelling LW by changing the local modelled weather, but a change in the surface flux is a prerequisite for this to take place. Hence the ice heat capacity should not prevent the separation of causes of the surface flux differences.

Using equation (13) we can separate the ice thickness-growth feedback from the different atmospheric forcing parameters at any point in model space and time. For a given model, at a point (**x**,t), the dependence of net surface flux on any component

of $F_{atmos-ic}$ under freezing conditions is $\sum_{cat=1}^{5} \frac{a_{ice}}{1+B_{up}R_{ice}}$, the 'ice thickness scale factor', which tends to $\sum_{cat=1}^{5} a_{ice}$ in the limit of thin ice and 0 in the limit of thick ice. In this way, given a model difference in downwelling LW of -50 Wm$^{-2}$ (for example), we multiply this by the model mean ice thickness scale factor to estimate the difference in net surface flux induced. Under thicker ice conditions, a given difference in atmospheric forcing will tend to induce a smaller difference in net surface flux, and hence a smaller difference in ice growth.

Given full information about the modelled ice thickness distribution, we can also use equation (13) to diagnose the effect of
       the thickness-growth feedback itself. At each model point ($\mathbf{x}$,t) this effect is represented by the sum over all categories of the
       effects of differences in category ice area and thickness. For each category, the rate of dependence of surface flux on
       category ice area is $1/(1 + B_{up}R_{ice})$, while the rate of dependence on category ice thickness is

$$-B_{up}a_{i-cat}/\left(k_i\left(1 + B_{up}R_{ice}\right)\right)^2.$$

Hence we can calculate surface flux difference induced by differences in atmospheric forcing ($F_{atmos}$), snow thickness, and
       in the ice thickness distribution, at each point in model space and time, and average these over the Arctic Ocean region and
       the period 1980-1999 to understand the effect of the evolving differences over the ice freezing season (Figure 7). For all
       three pairs of models, the effect of the ice thickness distribution (represented by the sum of the individual category terms)
       greatly outweighs the effects of differential atmospheric forcing, and of differences in snow cover. For example, when

comparing UKESM1.0-LL to HadGEM2-ES (Figure 7a) the total ice thickness-induced surface flux difference achieves a
       maximum of 15 Wm$^{-2}$ in November, representing a weaker upwards surface flux, and hence reduced ice growth, in
       UKESM1.0-LL. $F_{atmos}$ contributes significant differences only at the beginning of the freezing season, being -9.4 Wm$^{-2}$ in
       September but near-zero from December onwards, while snow differences contribute a maximum of -0.3 Wm$^{-2}$ surface flux
       difference, in January.

When we examine spatial patterns of the different components (not shown), the ice thickness component is seen to be
       strongest in the Atlantic sector and the Siberian seas, being above 20 Wm$^{-2}$ in the early freezing season, showing that the ice
       thickness differences attenuate ice growth in UKESM1.0-LL most in these regions. This is because ice in both models tends
       to be thinner here, rendering surface flux more sensitive to ice thickness. Nevertheless, the ice thickness component is also of
       substantial size in the Central Arctic throughout the freezing season, reducing from ~15 Wm$^{-2}$ in October to ~5 Wm$^{-2}$ in

April. The atmospheric surface flux term $F_{atmos}$ is large and negative (below -20 Wm$^{-2}$) in the seas bordering the Atlantic
       Ocean and Siberia in October and November, but is otherwise small.

       Ice thickness distribution-induced differences dominate also when HadGEM3-GC3.1-LL is compared to HadGEM2-ES (not
       shown) and also when UKESM1.0-LL is compared to HadGEM3-GC3.1-LL (Figure 7c), but in this last case contributions
       across all thickness categories are similar rather than being dominated by very thin ice as was found in the previous

comparison.. Unlike for the summer analysis, the required data exists for all model ensemble members, and ensemble
       standard deviation is seen to be small compared to the ensemble mean ISF differences. Hence the model differences in ice
       thickness distribution contribute to a systematic difference in seasonal ice growth and melt between models.

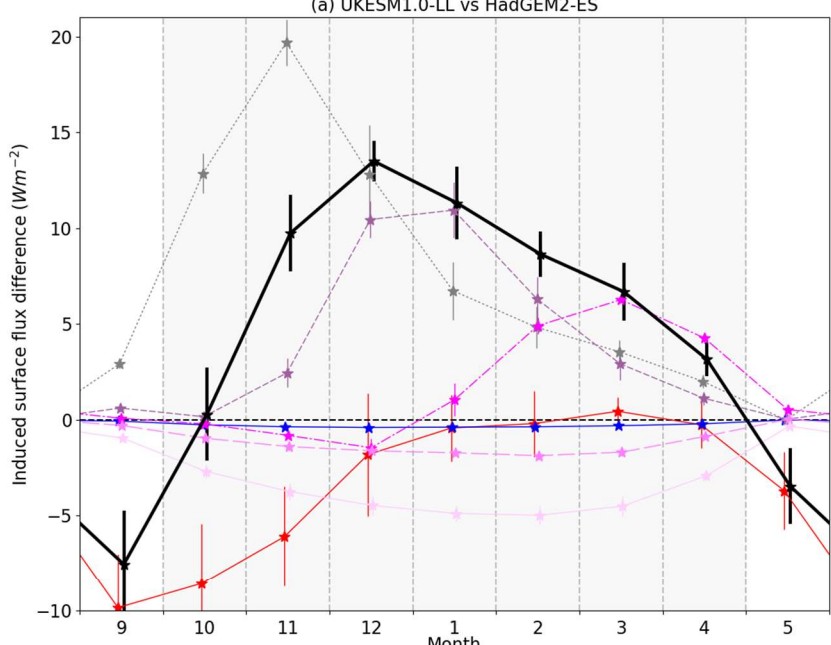

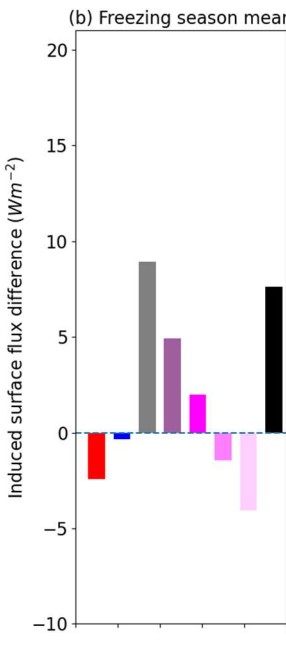

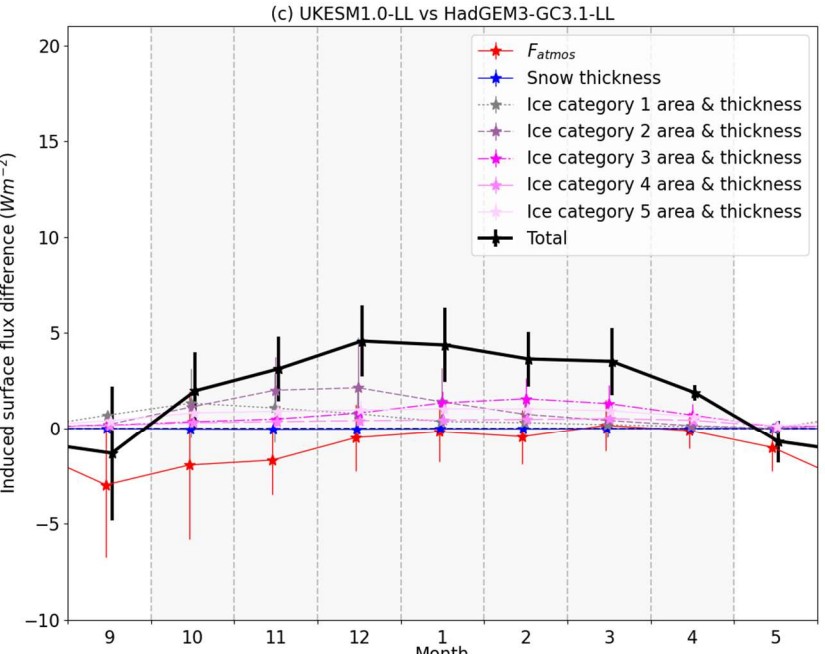

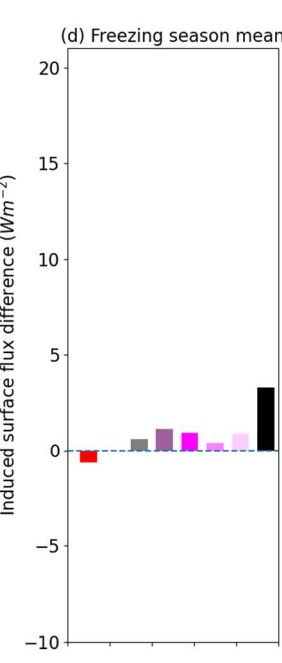

**Figure 7. Illustration of drivers of winter freezing differences, using the ISF framework. Shown are ISF freezing season differences due to atmospheric forcing, snow thickness and ice thickness for (a,b) UKESM1.0-LL relative to HadGEM2-ES; (c,d) UKESM1.0-LL relative to HadGEM3-GC3.1-LL. Showing September – May monthly evolution (left column) and freezing season mean (right column, October – April). For a) and c), the bars denote ensemble mean and standard deviation.**

These two approaches can be combined to provide a complete breakdown of the drivers of surface flux differences between pairs of models (Figure 8). We illustrate this with the UKESM1.0-LL – HadGEM2-ES comparison, performed using monthly means year-round as daily data is not available for the ice thickness distribution. Together with the ISF components, we plot the Arctic Ocean average ice export difference, and add this to the total ISF difference. The total ISF difference is negative May – September and positive from October – April. This indicates that model differences tend to drive a lower net down surface flux in the summer in UKESM1.0-LL (less ice melt) and a higher net down surface flux in the winter (less ice growth).

In summer, the ice area differences account for most of the negative ISF difference except in June. In this month, the ice area difference contributes only -13.2 $Wm^{-2}$ of the total -29.2 $Wm^{-2}$. The remaining negative ISF difference is accounted for by snow area albedo effect (-7.8 $Wm^{-2}$), surface melt onset (-4.5$Wm^{-2}$), and meltpond parameterization (-2.6$Wm^{-2}$), with an opposing effect from the snow area parameterization (4.4 $Wm^{-2}$). In addition, there are opposing contributions from the downwelling SW (6.8 $Wm^{-2}$) and downwelling LW (-9.8 $Wm^{-2}$) terms which largely cancel out.

In winter, the ice thickness distribution contributes almost the entire positive ISF difference, except in October and November, when there is a substantial negative term from the downwelling LW (-5.0 $Wm^{-2}$, comparing to 12.4 $Wm^{-2}$ from the ice thickness distribution over these two months). In both summer and winter, the ice export contributes a small positive flux equivalent to net loss of ice (2-3 $Wm^{-2}$), indicating more export from the Arctic Ocean in UKESM1.0-LL relative to HadGEM2-ES, consistent with the ice in UKESM1.0-LL being thicker.

Viewing the year as a whole, the ice growth/melt differences are almost all driven by differences in the ice state (area and volume) itself, with other drivers being important mainly during the early summer. The implications of this are discussed in Section 5 below. The other model comparisons are qualitatively similar (not shown): only when comparing UKESM1.0-LL to HadGEM3-GC3.1-LL in the melting season does a variable other than ice area or volume dominate the total ISF (snow thickness albedo effect). The ice area ISF difference in the melting season can be identified with the surface albedo feedback; the ice thickness ISF difference in the freezing season the thickness-growth feedback.

Finally we note that the total ISF difference is qualitatively consistent with the difference in net radiation between the models, and also with the difference in ice growth/melt, shown in Figure 6 for comparison. This gives confidence that the ISF framework is producing meaningful results. For example, from October – April the surface flux differences imply 58cm less ice growth in UKESM1.0-LL than in HadGEM2-ES; from May – September, they imply 75cm less ice melt. This

compares to a total growth/melt difference of 50cm. The difference in summer melt is likely overestimated in the ISF framework because some of the ISF differences occur over regions of already-melted ice, where they would affect the heating of the surface ocean layer, rather than the ice volume balance. This is consistent with the biggest difference between the total ISF and the ice growth/melt being in August, and the net radiation difference being closer to the total ISF than to the ice growth/melt in July and August.

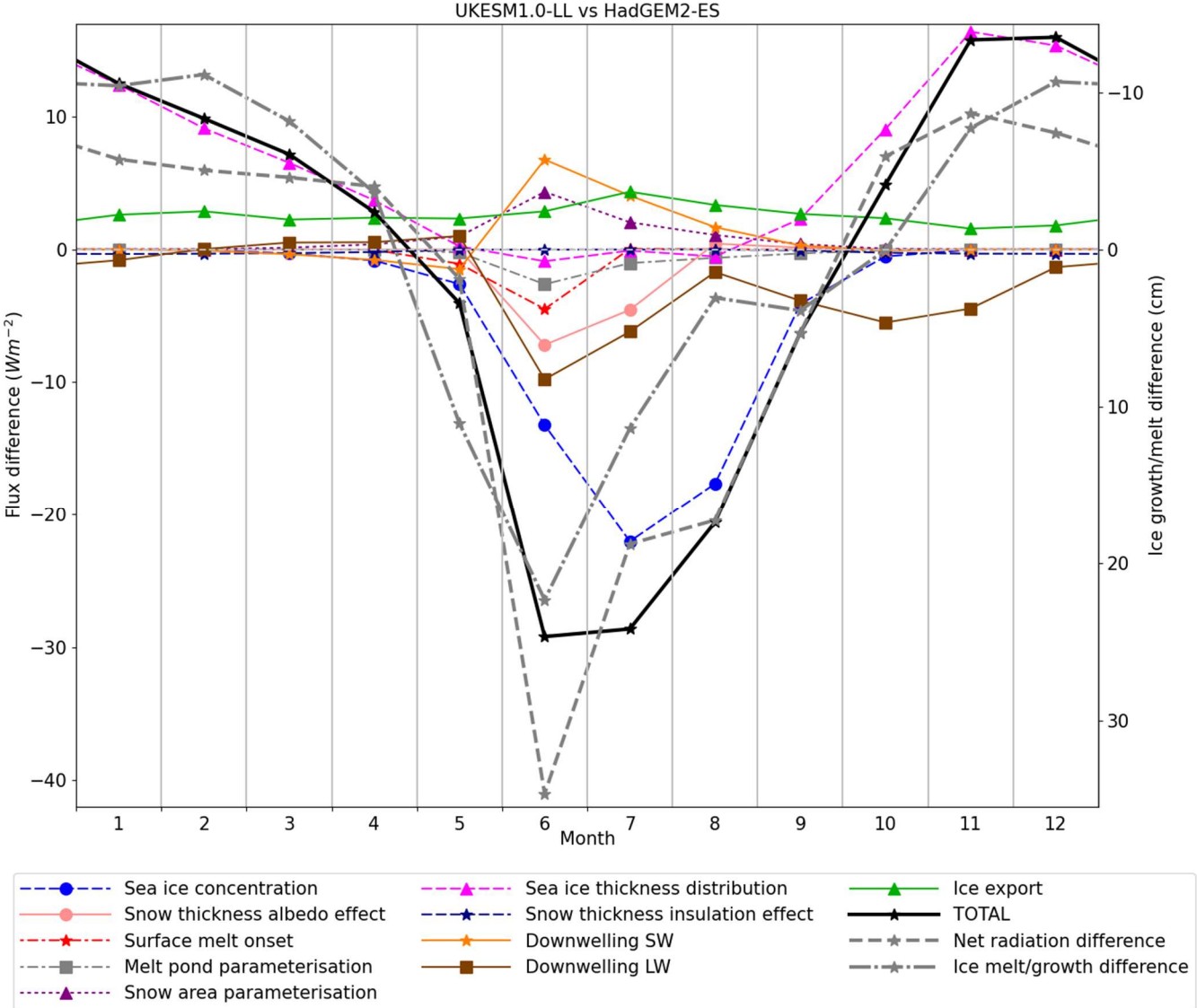

**Figure 8. ISF differences compared between pairs of models over the year as a whole, for UKESM1.0-LL relative to HadGEM2-ES. Averages are taken over the 1980-1999 period, and the Arctic Ocean region. The grey lines show, for comparison, the total net radiation difference and the ice growth/melt difference. All values are plotted in Wm$^{-2}$ (left**

**axis) while the right axis shows the equivalent monthly ice growth/melt difference in cm, assuming ice density of 917 kgm$^{-3}$ and latent heat of fusion $3.35 \times 10^5$ Jkg$^{-1}$.**

## 5 Discussion

In this section, we discuss the extent to which the ISF framework is able to explain the differences between the sea ice states
of UKESM1.0-LL, HadGEM3-GC3.1-LL and HadGEM2-ES. In accordance with the conceptual picture laid out in Section 3.2, we continue to assume that the surface flux is the primary driver of the sea ice melt/growth, and that mechanisms causing differences in surface flux cause proportionate differences in ice melt/growth.

The ISF framework shows that for the assessed models, the differences in seasonal ice growth/melt between models are driven almost entirely by differences in ice area and thickness. In a proximate sense however, the ice area and thickness –
collectively representing the ice volume – are driven by the seasonal ice growth/melt in turn. Because of this it is helpful to consider the ice volume, and the seasonal ice growth/melt, as a simple coupled system, with points of equilibrium where seasonal ice growth and melt are equal. Given an initial equilibrium, changes to an external variable such as downwelling radiation, or snow cover, lead to an initial change in annual growth and/or melt. This in turn induces a change in the ice volume (i.e. in the ice area or thickness), inducing a further change in the growth or melt via the effect of the surface albedo
feedback, and/or thickness-growth feedback, on the surface flux. The chain of causality continues until ice growth and melt are once again in balance, and a new equilibrium is reached.

Therefore the ISF differences not due to ice area and thickness show the proximate causes of the differences in the whole ice volume – seasonal ice growth/melt coupled system. For example, comparing UKESM1.0-LL and HadGEM2-ES, when the ice thickness and area contributions are excluded the largest remaining ISF differences are from the variables affecting
surface albedo in early summer: surface melt onset, melt pond parameterisation, snow thickness and snow parameterisation (contributions from downwelling SW and LW radiation largely cancel each other out). Collectively, they account for far less ISF difference than the ice area term. Yet they likely account for a major part of the difference between the ice volume – seasonal ice growth/melt systems of UKESM1.0-LL and HadGEM2-ES, because the ice area and ice volume differences can be traced to the initial reduced ice melting they trigger.

More generally, the 'external' variables – such as downwelling radiation, snow cover and surface melt – set the parameters of the ice volume - seasonal ice melt/growth coupled system. In effect, they determine the climate in which the sea ice will find an equilibrium volume, with ice melt and growth equal. This can be visualised by calculating ice melt and ice growth curves, as a function of annual mean ice thickness, with the simple model used in Section 4 (Figure 9), forced with the Arctic Ocean average surface radiative fluxes from each model. For each model, and for the real world, the external variables

determine the relationship between ice thickness, and ice growth and melt: an equilibrium ice thickness lies where the two curves meet.

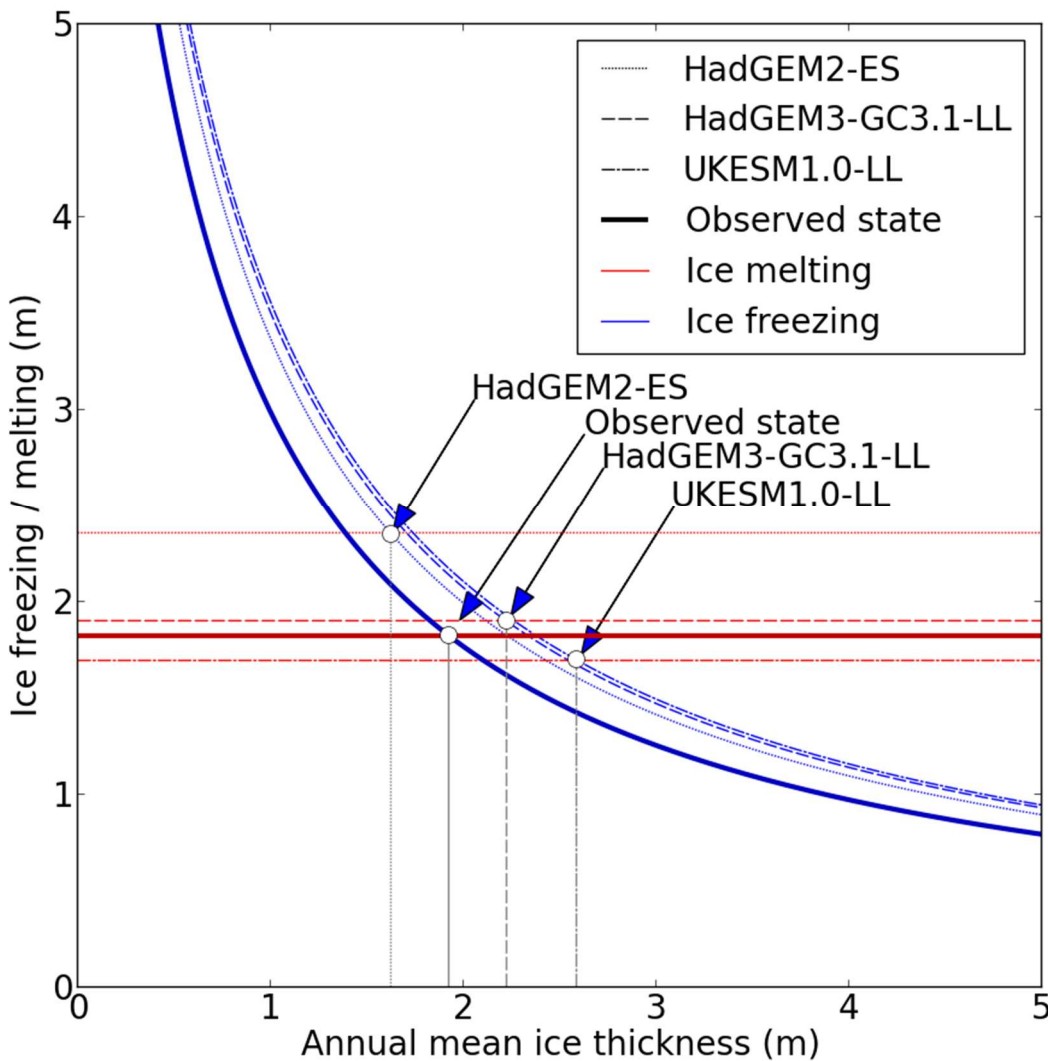

**Figure 9. An illustration of idealised ice thickness-ice growth (blue line) and ice thickness-ice melt (red line) relationships in the evaluated models and in observations, as produced by the ISF parameterisation. The graph**
**demonstrates how the ice growth and ice melt curves determine equilibrium ice thickness in each model climate.**

This conceptual picture accurately reproduces the qualitative differences between each model, and the model biases. All three models share similar ice growth curves, biased high relative to the real world: this reflects the similar low downwelling LW biases. The differences between models are caused by the ice melt curves. For example, the equilibrium ice thickness of HadGEM2-ES is much lower than that of UKESM1.0-LL because this is the only way to achieve the greater ice growth required to balance the ice melt differences.

The impact of external drivers on the sea ice varies greatly by time of year, because of this interplay between ice thickness and ice growth/melt. On the one hand, the effect of surface flux biases during the freezing season is diminished by the thickness-growth feedback, particularly early in the season. Differences created are reduced as the freezing season proceeds, because thin ice grows more quickly than thick ice. On the other hand, the effect of surface flux biases during the melting season is enhanced by the surface albedo feedback, particularly early in the season. Differences created increase as the season progresses, as thinner, warmer, less extensive ice has a lower albedo than thicker, colder, more extensive ice. Hence small differences in forcing can have a large effect in the late spring and early summer, whereas small differences during the freezing season will tend to only have a small impact. This is consistent with the prediction of DeWeaver et al. (2008) that that sea ice state is more sensitive to surface forcing during the ice melt season than during the ice freezing season.

A particularly useful aspect of the ISF analysis is that the direct effect of model parameterisation changes can be quantified, namely the change to an explicit melt-pond scheme and the change of snow area parameterisation. In each case, the impact on the surface flux, although small compared to the melt onset and snow thickness terms (let alone the ice area term), is not negligible, and because the impact is felt in the early summer likely has a significant impact on the sea ice state. This offers a useful perspective on the importance of sea ice model improvements versus model forcing. The effect of such model improvements may be small, but could still have significant effects, particularly if the effects are concentrated in the crucial late-spring to early-summer time of the year.

## 6 Conclusions

The models HadGEM2-ES, HadGEM3-GC3.1-LL and UKESM1.0-LL have been compared using a systematic framework (the ISF, or induced surface flux framework) to quantify the impact of differences in individual model variables on differences in modelled sea ice melt and growth. Of the three models, UKESM1.0-LL displays the highest annual mean ice thickness and September ice area, and the least annual ice growth/melt, while HadGEM2-ES displays the lowest annual mean ice thickness, and the most annual ice growth/melt. These are consistent with differences in surface fluxes, with UKESM1.0-LL (HadGEM2-ES) displaying the lowest (highest) net SW flux during the summer and the highest or least negative (lowest or most negative) net LW flux during the winter. The ISF framework shows that the major part of the difference in surface flux between each pair of models is caused by differences in the ice state itself, via the surface albedo feedback in the summer and the thickness-growth feedback in the winter. In this way, it demonstrates how closely coupled the seasonal ice growth/melt is to the ice volume.

The remainder of the surface flux differences can mostly be attributed to variables influencing surface albedo in June. The snow thickness and melt onset terms, in particular, drive a higher surface albedo, and hence a lower surface flux, in both CMIP6 models relative to HadGEM2-ES, and in UKESM1.0-LL relative to HadGEM3-GC3.1-LL. These represent the proximate causes of model differences in the ice volume – seasonal ice melt/growth coupled system. Small differences in surface flux at this time of year are magnified by the surface albedo feedback into much larger differences in ice melt later in the season, as represented by the ice area ISF term.

The ISF framework focuses attention on model processes most likely to be implicated in driving inter-model spread, helping inform future model development. For the models examined in this study for example, small differences in the surface energy budget in the early summer appear to lead to very large differences in the sea ice state. Differences in snow area and melt onset in particular are implicated. This underlines that these are likely to be particularly important variables to model correctly in order to reduce sea ice state errors.

The framework also allows the proximate effect of some specific model improvements on seasonal ice growth/melt to be measured directly. The effect of the explicit melt-pond scheme and new snow parameterization of the CMIP6 models on the sea ice volume balance, relative to other model drivers, is shown to be small but non-negligible, precisely because their greatest effect on the surface flux occurs in the early summer, when small differences in ice melt can lead to much greater differences later in the season.

**Code availability**

Code used to calculate the ISF contributions, and to plot all figures in the paper, is archived at https://zenodo.org/record/5675109.

**Data availability**

Sea ice and surface flux data for HadGEM3-GC3.1-LL and UKESM1.0-LL can be downloaded from the CMIP6 archive at https://esgf-index1.ceda.ac.uk/search/cmip6-ceda/ under the model labels HadGEM3-GC31-LL and UKESM1-0-LL. Data for HadGEM2-ES can be downloaded from the CMIP5 archive at https://esgf-index1.ceda.ac.uk/search/cmip5-ceda/.

**Author contribution**

The ISF framework was designed and carried out for the case study models, along with the model evaluation and comparison in Section 3, by Alex West. The paper was written in its final form by Alex West with assistance and advice from Ed Blockley and Mat Collins.

**Competing interests**

The authors declare that they have no conflict of interest.

**Acknowledgements**

This work was supported by the Met Office Hadley Centre Climate Programme funded by BEIS and Defra, and the European Union's Horizon 2020 Research & Innovation programme through grant agreement No. 727862 APPLICATE.
MC was supported by NE/S004645/1. The authors would like to thank the two anonymous reviewers for their helpful feedback.

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
