# Peer review of "Understanding model spread in sea ice volume by attribution of model differences in seasonal ice growth and melt"

_The Cryosphere, 2021_

## Referee Comment (RC2)

**A review on "Understanding model spread in sea ice volume by attribution of model differences in seasonal ice growth and melt"**

The study evaluates the three UK CMIP models performance in reproducing the Arctic sea ice volume seasonal cycle and discusses the inter-model spread using a simple surface energy balance model as a diagnostic tool. The subject of the study is certainly important as the large spread between the coupled climate models concerning the Arctic sea ice volume and extents results in the uncertainty of future climate projections in the Arctic and lower latitudes. Understanding such a spread is crucial for the identification and/or development of more adequate models. A simple approach to explain the model spread is presented by the authors. It consists in using an idealized representation of the sea ice bulk and of the surface energy balance to provide a reference or a framework for the analysis of the more complex model results. It is shown that such an approach can provide some useful estimates of the sensitivity of the net surface heat flux on model variables allowing one to draw important conclusions on the relative role of various factors affecting the seasonal cycle of the sea ice cover. In general, the paper is well written and represents a significant input in the research in this area, and the subject is in the scope of The Cryosphere. The paper can be published after minor revision.

**General comments:**

1. The authors propose to use a one-dimensional heat balance model. Obviously, in such a box-type or bulk model the sea ice dynamics is neglected. At the same time, we know that the sea ice volume shows significant geographical variability across the Arctic related not only to the variability of the surface heat budget terms, but also associated with the sea ice drift and deformation. Thus, one can expect that changes in the models physics and resolution can affect the sea ice dynamics and it can affect the sea ice volume and contribute to the spread between the models. The authors do not discuss such issues at all. How well do the considered models reproduce the 2D sea ice dynamics? Is there any spread between the models with respect to the sea ice dynamics? Can we expect that different representation of the sea ice dynamics, e.g. amount of the sea ice transport through Fram Strait, can affect the simulated sea ice volume and its annual cycle?
 I understand, that to some extent, averaging over the Arctic ocean solves this problem. However, this should be discussed in more detail. For example, it might be important at which step and how the averaging is done. As far as I understood, the simple model is used at each grid node and then the obtained results are averaged over the Arctic ocean. But at each grid node the advective flux of the sea ice volume is not negligible especially in some regions. Thus, the single-column approach has to be better justified.

2. The authors obviously neglect the heat flux from the ocean to sea ice which is especially important in the Atlantic sector of the Arctic. The authors should discuss the magnitude of this term in relation with the other terms in their simple model.

3. My last comment is related to the applicability of a simple model that the authors use as a diagnostic tool. Obviously, various models can describe the sea ice thermodynamics differently than it is done in such a simple model. The actual sensitivity of the net flux in a particular sea ice model to the model variables can differ from model to model and would also depend on the considered time scale. How large can we expect such differences to be?

**Specific comments**

*Line 193:* To summarise, the weaker summer ice melt of the CMIP6 models relative to HadGEM2-ES is driven by a smaller upwelling SW flux from June – August.
It can be the other way round – weaker melt results in a more negative SW flux due to larger sea ice area. How is it possible to identify the cause?

*Section 3.3. Variables influencing surface albedo* – I suggest to explicitly write the albedo parameterizations used in the models, so that the reader can clearly see what are the variables influencing albedo.

*Equations 1 and 2* – variables have to be explained

*Lines 220-224:* "Despite the substantially higher snow thicknesses in HadGEM3-GC3.1-LL and UKESM1.0-LL, the increase in ice area in the newer models is muted…."  It is not easy to follow because there is no reference formula for albedo. How does albedo depend on the snow thickness and ice area? It is not clear

*Lines 254-255:* It is assumed that the net heat flux is a function of some model variables which are independent of heat flux. But this is not true on the considered time scales. Obviously, albedo and melt pond fraction would depend on the net surface heat flux already on a weekly and monthly time scales. Does it result in a limit of applicability of this assumption?

*Equation 3* – superscripts MODEL1 and MODEL2 are not visible.

*Line 261:* I suggest to write explicitly how the ice volume balance is related to the surface heat flux. I wonder why the ice volume tendency is omitted in the simple model.

*Equations 5 and 6* – I suggest different letters for the variable a_melt and the area fractions a_i. Maybe, use capital A for the area fractions, otherwise it is confusing.
Equation 6 – Fsw-net, t is missing
What is the exact definition of a_melt in Section 3 and how is Equation 6 obtained? It is hard to follow.

*Line 275:* We can use this equation – specify which equation

Obviously, *Equation 7* cannot be used for category zero (open water)

*Line 351:* How is it linearized and what is Bup?

*Lines 355-357:* First it is stated that Fatmos-ice does not depend on the surface temperature. Next, Fatmos-ice is identified as sum of SW net, LW down and turbulent fluxes. Obviously, turbulent fluxes do depend on surface temperature. It can be argued that LW down also depends on surface temperature on the time scale of the atmospheric boundary layer adjustment (which is not large), because the near-surface air temperature over sea ice is coupled to surface temperature.

*Figure 7 and lines 445-450:* It should be better explained how the curves in Figure 7 are obtained. Ice melt and ice growth are not described by the model in Section 4. Such terms are simply missing. So it is not clear at all how Figure 7 is obtained.

***Line 479:*** modelled sea ice and growth (??)

---

## Author Response (AR1)

**Author's response accompanying revised version of 'Understanding model spread in sea ice volume by attribution of model differences in seasonal ice growth and melt'**

This response is set out in the following way. Firstly, updated responses to the two anonymous reviewers are given in turn. Secondly, a complete list of changes to the document is presented.

**Updated response to first anonymous review of 'Understanding model spread in sea ice volume by attribution of model differences in seasonal ice growth and melt'**

In the following response, the reviewer's comments (italic font) and our original response (normal font) are shown in black. Our updated response, describing changes made to the paper in association with each respective comment, is shown in green.

***Review 1***

*The authors analyse how differences in three climate models contribute to difference in the modelled sea ice. They apply the ISF method to break out contributions of individual components to drivers of melt and freeze throughout the year. Overall the manuscript is well written, clear, and has sound methodology. Well done! My concerns are primarily related to clarification and minor in scope.*

We thank the reviewer for their kind comments about our study. We apologise for the time taken to produce this response. Some of the issues raised by Reviewer 2 required detailed consideration of the timescales of aspects of ice thermodynamics, and of the atmospheric boundary layer, before we felt an adequate reply could be submitted. In addition, three weeks were lost due to COVID in the family. We wished to ensure maximum consistency between our responses, and therefore submit them at the same time.

***Specific comments***

- *Line 110, remove "and" after Wang et al.*

This will be done. This has been done.

- *Line 197: I would recommend rewording "thicker ice in HadGEM3-GC3.1-LL and UKESM1.0-LL causing a colder surface temperature, and less heat loss to space, than is the case in HadGEM2-ES" to: "compared to HadGEM2-ES, the two CMIP6 models have thicker ice which leads to a colder surface due to reduced heat conduction through the ice, and the colder surface results in less longwave radiative loss to space."*

Yes, this is much clearer and we will amend this as suggested. This has been amended as suggested.

- *Line 216: You haven't cited or mentioned CICE yet. You may want to expand a bit about why or how the ice models are different.*

We have added information in section 2 to clarify that the two CMIP6 models use CICE, and specified the configuration. We have added information in section 2 to clarify that the two CMIP6 models use CICE, and specified the configuration, with a reference to Ridley et al. (2018).

- *Figure 3:*
    - *Could you plot surface albedo differences to show the net spatial impacts of all components?*

This step would be quite similar to the main ISF analysis – quantifying the impacts on surface flux of all the components, the main difference being that it involves quantifying the impact on surface albedo, i.e. going back one step in the causal chain. We will consider how best to work this in, whether in a separate figure, additional panels, or in a supplementary figure.

Taking any sort of average of surface albedo values involves a subjective judgment: take a simple arithmetic mean, or first weight by downwelling SW? We would probably adopt the first approach as it will show the immediate effect of the various differences more clearly, and the second approach would produce an answer identical (up to scaling by a constant) to the main ISF analysis.

For completeness, we have added an ice area row to Figure 3 (now Figure 4), considering this to be necessary in order to properly address both this and the following suggestion by the reviewer.

We have estimated components of surface albedo difference due to ice area, snow thickness, snow area parameterisation, surface melting tendency and meltpond parameterisation (using UKESM1.0-LL relative to HadGEM2-ES as an example). The sum of the components is very similar to the actual surface albedo difference. Because this analysis works well as an 'introduction' to the main ISF analysis, we have presented it in a new Figure 5 and summarised it in the text, drawing attention to how the ISF analysis proceeds naturally from considering the main shortcoming (surface albedo differences can't be averaged over time and space).

    - *It might be nice to compare with observations here, where appropriate. The different models have hugely different ice fractions and melt pond fractions. Which are most reasonable given observations?*

This is a good idea and we will try to work this in, whether in additional panels or by means of text description. Some of the variables plotted here are quite poorly observed (particularly snow fraction) but it should be possible to say something about the melt pond and ice fractions.

We have added observational datasets of ice area, surface melting tendency and meltpond fraction to Figure 4, commenting on how the model differences relate to these observations in the text.

- *Equation 2: instead of MODEL it says MODE. Same two lines above.*

In fact these should read MODEL1, MODEL2, MODEL1 and MODEL2, and seem to have been truncated somehow before publication, which we should have noticed. We will make sure that these are correct in any future published revisions. We have confirmed that these are and remain MODEL1, MODEL2, MODEL1 and MODEL2, and will pay particular attention to their appearance after editing.

- *Equation 5: At first it is nearly impossible to tell the difference between a for area and alpha for albedo. Could Area be changed to $A_i$ or bolded to make this clearer? Same for lines below.*

This is also a good idea, which Reviewer 2 mentioned. Thank you. All instances of 'a' denoting area have been changed to 'A'.

- *Line 282: Please clarify how bare ice fraction is found.*

Bare ice fraction is one minus the sum of the other terms. This will be stated in the text. This has been stated in the text.

- *Line 285: If this equation is relevant, may want to number it. Also, please define the albedo of the ocean. Is the albedo over different surface types output directly?*

This equation will be moved out of the paragraph and numbered. The albedo of the ocean will be defined.

In fact, the albedo of the ocean is subject to relatively complex parameterisations in all three models; this is now indicated in the text, with references given. Rather than attempt to reproduce this, we calculated ocean albedo in the Arctic from model diagnostics (downwelling and upwelling SW), and set ocean albedo in our simple model to be 0.07, a rounding of the modal values in all three models to two decimal places. This was slightly different to the 0.06 we originally used but the results are not qualitatively affected by this change. We have described our methodology in the text.

- *Line 304-306: Did you verify that the answer is similar by using either/both CMIP6 models and comparing ensemble mean to the individual ensemble members?*

No, and this would be a good idea. We will carry out this comparison and report on this in a revision.

In the revised version of the paper, we use all four ensemble members for the UKESM1.0-LL vs HadGEM3-GC3.1-LL comparison, and show ensemble standard deviation as shaded regions in the daily graph (and error bars in the melt season mean). This does in fact result in a qualitative change in results: over the melt season as a whole, the snow thickness and ice area components are now comparable in magnitude, instead of the snow thickness being somewhat larger. This occurs because the ice area difference is somewhat larger in the second and third ensemble members than in the first and fourth. We have updated the text with reference to this.

- *Figures 4,5,6:*
  - *These figures are really nice and clear, but I had a lot of trouble with the colors. Please modify the colors to improve readability, change dash patterns, or bold particular lines of relevance.*

Thank you, this is useful feedback. We will try to use a mixture of linestyles and colours to distinguish the variables.

We have altered the colours used on these figures (now Figures 6, 7 and 8) where it seemed colours previously used were too similar, and have used differing linestyles to try to enhance the clarity further.

  - *I think monthly mean figures of the spatial difference contributions might be useful as a supplementary figure to see which components dominate in different regions.*

This is a good idea and we will look at spatial patterns of the ISF differences next. The spatial patterns observed will probably be described in the main text, though as you say any associated figure can be supplementary (unless there is anything really striking).

We have examined spatial patterns of the various ISF terms in both the melting and freezing season analysis, and have summarised these in the text.

- *Line 328: Why is the snow thickness so different between these models?*

We are not sure. The baseline climate in UKESM1.0-LL is considerably colder in the period in question, so snow accumulation during the freezing season may be greater (though this association is obviously by no means automatic, as one would also expect total precipitation to be lower in the colder model). We will look into this further by examining seasonal and spatial patterns of snowfall, but a more in-depth analysis (e.g. by analysing the role of atmospheric circulation differences) would probably be beyond the scope of this study.

We have examined snowfall over sea ice in UKESM1.0-LL and HadGEM3-GC3.1-LL, and found very little difference. The difference in snow thicknesses during the early melt season appear to arise entirely from the different phasing of surface melting in the two models. We note that the size of the snow thickness component in the UKESM1.0-LL vs HadGEM3-GC3.1-LL comparison is similar in magnitude to that of the UKESM1.0 vs HadGEM2-ES comparison. In the second comparison, however, the ice area term is much larger. Hence it is the ice area term that appears to be the anomaly here. We conjecture that this arises from the interplay between ice thickness and ice area: more ice area is lost during a melting season at lower thicknesses. Hence the ice area term naturally plays a proportionally larger role in differences wrt HadGEM2-ES (with the thinnest ice) than wrt HadGEM3-GC3.1. We have attempted to summarise this reasoning in the text.

- *Line 379: I would recommend rewording "differences in the thicker categories contribute much more towards the total" to: "contributions across all thickness categories are similar rather than being dominated by very thin ice as was found in the previous comparison."*

Yes, this is much clearer and will be amended as suggested. This has been amended as suggested.

**Updated response to second anonymous review of 'Understanding model spread in sea ice volume by attribution of model differences in seasonal ice growth and melt'**

In the following response, the reviewer's comments (italic font) and our original response (normal font) are shown in black. Our updated response, describing changes made to the paper in association with each respective comment, is shown in green.

*Review 2*

*A review on "Understanding model spread in sea ice volume by attribution of model differences in seasonal ice growth and melt"*

*The study evaluates the three UK CMIP models performance in reproducing the Arctic sea ice volume seasonal cycle and discusses the inter-model spread using a simple surface energy balance model as*

*a diagnostic tool. The subject of the study is certainly important as the large spread between the coupled climate models concerning the Arctic sea ice volume and extents results in the uncertainty of future climate projections in the Arctic and lower latitudes.*

*Understanding such a spread is crucial for the identification and/or development of more adequate models. A simple approach to explain the model spread is presented by the authors. It consists in using an idealized representation of the sea ice bulk and of the surface energy balance to provide a reference or a framework for the analysis of the more complex model results. It is shown that such an approach can provide some useful estimates of the sensitivity of the net surface heat flux on model variables allowing one to draw important conclusions on the relative role of various factors affecting the seasonal cycle of the sea ice cover. In general, the paper is well written and represents a significant input in the research in this area, and the subject is in the scope of The Cryosphere. The paper can be published after minor revision.*

We thank the reviewer for their kind comments. We apologise for the time taken to produce this response. The reviewer asked a number of perceptive questions about timescales and the relevance of various physical processes. Constructing a satisfactory reply to these required detailed consideration of sea ice thermodynamics, and of atmospheric boundary layer processes. This in turn involved further examination of model data, of the literature, and of properties of the heat equation under changes in surface forcing, which we have attempted to describe in brief in our response. In addition, three weeks were lost due to COVID in the family.

*General comments:*

*1. The authors propose to use a one-dimensional heat balance model. Obviously, in such a box-type or bulk model the sea ice dynamics is neglected. At the same time, we know that the sea ice volume shows significant geographical variability across the Arctic related not only to the variability of the surface heat budget terms, but also associated with the sea ice drift and deformation. Thus, one can expect that changes in the models physics and resolution can affect the sea ice dynamics and it can affect the sea ice volume and contribute to the spread between the models. The authors do not discuss such issues at all. How well do the considered models reproduce the 2D sea ice dynamics? Is there any spread between the models with respect to the sea ice dynamics? Can we expect that different representation of the sea ice dynamics, e.g. amount of the sea ice transport through Fram Strait, can affect the simulated sea ice volume and its annual cycle? I understand, that to some extent, averaging over the Arctic ocean solves this problem. However, this should be discussed in more detail. For example, it might be important at which step and how the averaging is done. As far as I understood, the simple model is used at each grid node and then the obtained results are averaged over the Arctic ocean. But at each grid node the advective flux of the sea ice volume is not negligible especially in some regions. Thus, the single-column approach has to be better justified.*

Our simple model neglects the effects of ice dynamics by construction, because it is designed to diagnose the causes of differences in *surface heat flux*, rather than in ice volume tendency directly. At the end of section 3.1 (paragraph beginning at line 161) we discuss the link between surface heat flux and ice volume tendency. We will try to make this discussion more prominent, because it is key to the setup of the simple model, and to the issues around ice dynamics and oceanic heat convergence, and because the reviewer asks two other questions about ice volume tendency below to which it is also relevant.

Basically, there are three processes affecting ice volume tendency (or ice growth/melt as we describe it in our study): the surface heat flux, the ice divergence, and the basal ice-ocean heat flux. Our study is principally designed around diagnosing the causes of differences in the first of these processes, the surface heat flux. In line 162 we state the Arctic Ocean average ice divergence in the three models, noting that it is small. However, our study would probably be more complete if we added an explicit 'ice divergence' line to Figure 6. To be clear, this would not be part of the simple model, which estimates the effects of various factors on surface flux differences. Rather, it would complement the induced surface flux differences, representing an additional factor impacting ice volume tendency. We suspect that it would appear quite flat on Figure 6, and appear near zero year-round on the given scale, but it would be useful to see this.

The treatment of the basal ice-ocean heat flux is more complex (see below).

We have added a new subsection 3.2, between the ice volume and surface radiation evaluation, specifically to justify the treatment of the surface flux as the principal driver of the ice melt/growth, with a new schematic figure. This includes a discussion of ice export. It also includes a discussion of the issue raised in the reviewer's next comment, the relationship between ice-ocean heat flux, oceanic heat uptake and oceanic heat convergence. We have also added an ice export term on Figure 6 (now Figure 8) as this directly affects the relationship between surface flux and ice melt/growth.

*2. The authors obviously neglect the heat flux from the ocean to sea ice which is especially important in the Atlantic sector of the Arctic. The authors should discuss the magnitude of this term in relation with the other terms in their simple model.*

This is a good point, though it's more accurate to say that we neglect the oceanic heat convergence rather than the ocean-ice heat flux.

The ocean-ice heat flux has two sources: via oceanic heat convergence, or via exchange of surface heat flux in ice-free areas (e.g. polynyas, the marginal ice zone). Studies show (e.g. Serreze et al. 2007) that the ocean-ice heat flux in much of the Arctic displays a pronounced seasonal cycle, being near-zero in winter (except near the Atlantic ice edge, see below) and significant in size only in summer, particularly late summer. There is plentiful evidence that the ocean heat energy released in late summer derives from direct solar heating (i.e. the surface heat flux) rather than from oceanic heat convergence, both in the real world (McPhee et al., 2003; Perovich et al., 2008) and in models (Steele et al., 2010; Keen and Blockley, 2018).

Hence, in order to crapture the first-order effect of the ocean-ice heat flux, it is sufficient to account for direct solar heating of the ocean – which our model does in fact do. This is because we estimate the gridbox mean surface heat flux – the surface heat flux over all surface types, not just ice. Differences in solar heating of the ocean therefore show up as surface flux differences in our model. This has drawbacks – not all solar heating is going to be converted into ice melt, for example – but it means that a first-order driver of ice volume tendency is captured.

The oceanic heat convergence represents an additional term in driving ice volume tendency, although we note that as above some of this heat can be released directly from the sea surface rather than affecting sea ice. In HadGEM2-ES, HadGEM3-GC3.1-LL and UKESM1.0-LL the Arctic Ocean average oceanic heat convergence is 4.4, 3.8 and 3.9 Wm-2 respectively, but most of this heat is

released very close to the Atlantic sea ice edge. We will include a discussion of the role of this term in our revised version.

As indicated above, this issue is discussed in our new Section 3.2.

*3. My last comment is related to the applicability of a simple model that the authors use as a diagnostic tool. Obviously, various models can describe the sea ice thermodynamics differently than it is done in such a simple model. The actual sensitivity of the net flux in a particular sea ice model to the model variables can differ from model to model and would also depend on the considered time scale. How large can we expect such differences to be?*

The reviewer is correct to point out that the representation of sea ice thermodynamics in our simple model is much more simplistic than is the case in the current generation of sea ice models (although it is quite similar to that of HadGEM2-ES, the CMIP5 model in our study). The most obvious example, to us, is the representation of the ice heat capacity: our simple model treats the ice as having no heat capacity, responding instantly to changes in surface forcing, whereas most (all?) CMIP6 ice models model ice heat capacity, with multiple layers with temperatures that respond on finite timescales to changes in surface forcing.

How would this affect the sensitivity of surface flux to the various variables considered? It's most relevant to the freezing season analysis, where we assume in equation (7) a uniform conductive flux through the ice, with the entire ice column responding instantly to a change in surface forcing. This is what actually happens in HadGEM2-ES, but in the two CMIP6 models the ice column would respond much more slowly. On a short timescale, the response of the surface flux to a large step change in e.g. downwelling LW would be representative of a thinner ice column (as only the top portion of the ice column would react quickly to the change in forcing). In other words, the surface flux would be *more* sensitive (on a short timescale) to a change in downwelling LW than is suggested by our method – because the damping effect of the ice thickness-growth feedback takes time to take effect.

However, there are several reasons why we do not think this is a major problem for our analysis. Firstly, this effect is weakest in the thinnest ice categories, which account for most of the heat loss (and hence *difference* in heat loss). In order to properly quantify the timescales at work here, we solved the heat equation for an ice column of thickness $h$ in the case of a sudden step change in energy flux at the top surface. This approach is described in more detail in an appendix to this response, but the solution is described by an infinite sum of decaying harmonics, the first of which decays the slowest and whose exponent therefore describes the timescale over which the ice column temperature profile approaches a new linear equilibrium. Using this, for each model ice category we can describe the range of e-folding timescales (Table R1):

| Category | Thickness range | Range of timescales over which slowest harmonic decays by 1/e |
|---|---|---|
| 1 | 0-0.6m | 0-21 hr |
| 2 | 0.6-1.4m | 21 hr – 3.5 days |
| 3 | 1.4-2.4m | 3.5 – 8.9 days |
| 4 | 2.4-3.6m | 8.9 – 18 days |
| 5 | 3.6m- | 18 days - |

*Table R1. Analytically-derived timescales of response to a sudden change in surface forcing, by ice category*

In other words, in the two thinnest categories the zero-layer approximation describes the surface flux response quite accurately after a few days. This is still a sufficiently short timescale for our purposes – and most of the surface heat flux variability comes from these categories.

Secondly, it's useful to view the different variables as acting 'instantaneously' on surface flux in our simple model because this helps to refine causality. In our simple zero-layer model, ice thickness and downwelling LW act instantly on surface flux, and this means that we can define differences in each variable as separately contributing a 'proximate cause' of the surface flux difference. The key here is that it's not necessary that each acts on surface flux instantaneously, just that they act more quickly than they act on each other. In the thickest ice categories, the full effect of a change in ice thickness or downwelling LW is realised slowly, over weeks or even months. But the result of this is that the full effect of a change of ice thickness on downwelling LW, or vice versa, is realised even more slowly. The thermal inertia slows the whole system down, not just the effects on surface flux. Because of this, even in the thickest ice categories we can still resolve the causality to an extent: changes in downwelling LW have to act on the surface flux before acting on the ice thickness. Hence we can still say that we are resolving the proximate cause of the surface flux difference.

Lastly, in reality the atmospheric forcing will display many changes, upward and downward, over the course of a freezing season. In general, the ISF error due to the thermal inertia effect will be greatest following a rapid change in downwelling LW in one model relative to the other. When averaged over longer timescales (such as monthly means), we would expect the effect of rapid downwards changes to largely cancel with the effect of rapid upward changes, meaning that the overall error is small.

The arguments here are likely too long to include fully in our study, but we will try to condense this into a brief discussion of this issue towards the end, and hope that we have answered the reviewer's question to their satisfaction.

For now, we have summarised this issue in a paragraph in Section 4, when the freezing season ISF analysis is described. This appeared to us to be the most natural place to describe this issue, as we justify here why the variables we are examining are 'quasi-independent', i.e. operate on the surface flux on a sufficiently short timescale.

**Specific comments**

*Line 193: To summarise, the weaker summer ice melt of the CMIP6 models relative to HadGEM2-ES is driven by a smaller upwelling SW flux from June – August. It can be the other way round – weaker melt results in a more negative SW flux due to larger sea ice area. How is it possible to identify the cause?*

The sentence after the one quoted by the reviewer is relevant here: 'This [the smaller upwelling SW flux] is likely caused by ice area differences in July and August (the surface albedo feedback), but in June other surface albedo drivers are responsible.'

Put simply, upwelling SW differences driving ice melt differences, and ice melt differences driving upwelling SW differences via ice area & surface albedo, are not mutually exclusive. Both processes are almost certainly in action here. Nevertheless the purpose of this section is to trace the ice melt/growth differences back to an immediate, proximate, driver, and the upwelling SW differences

are the obvious candidate. We note in the very same paragraph in which the upwelling SW differences are identified (beginning line 169) that ice area differences almost certainly explain the upwelling SW differences in July and August, but that in June they are not sufficiently large. This then motivates the surface albedo discussion in Section 3.3, and subsequently the whole ISF melt season analysis.

In summary, upwelling SW and ice melt differences almost certainly drive each other to a large extent; the question is to what extent do they drive each other, and at what times of the melting season, and this is one of the questions that our analysis addresses.

*Section 3.3. Variables influencing surface albedo – I suggest to explicitly write the albedo parameterizations used in the models, so that the reader can clearly see what are the variables influencing albedo.*

In analysing surface albedo differences, the first step is to consider differences in the area and albedo of the different surface types present in a grid cell. This is summarised by equation (5) in section 4, but this information may as the reviewer says be better placed here.

Following on from this, in analysing differences in the area and albedo of the different surface types, two parameterisations are relevant. The first is of snow area, for which the parameterisation are already quoted in the text. The second is of meltpond area, which is parameterised in HadGEM2-ES and whose formulation we will describe in greater detail. (In the CMIP6 models as stated, meltpond area is explicitly modelled and no parameterisation can be described). All other relevant variables are either explicitly modelled or are single parameters whose value will be stated.

We have moved equation (5) to the beginning of Section 3.3, and expanded the description of the variables influencing surface albedo. We have also described the HadGEM2-ES meltpond parameterisation in more detail when the meltpond variables are analysed in Figure 4.

*Equations 1 and 2 – variables have to be explained*

Apologies for this omission – variable definitions will be added. Variable definitions have been added.

*Lines 220-224: "Despite the substantially higher snow thicknesses in HadGEM3-GC3.1-LL and UKESM1.0-LL, the increase in ice area in the newer models is muted…." It is not easy to follow because there is no reference formula for albedo. How does albedo depend on the snow thickness and ice area? It is not clear*

This should have read 'increase in snow area' rather than 'increase in ice area', and will be corrected. The formulae for calculating snow area from snow thickness are given in equations (1) and (2) and hence the statement follows from these, and from Figure S1: the statement does not actually depend on the parameterisation of the surface albedo itself. 'Increase in ice area' has been corrected to 'increase in snow area'.

*Lines 254-255: It is assumed that the net heat flux is a function of some model variables which are independent of heat flux. But this is not true on the considered time scales. Obviously, albedo and melt pond fraction would depend on the net surface heat flux already on a weekly and monthly time scales. Does it result in a limit of applicability of this assumption?*

'… variables which are quasi-independent in the sense that while they affect surface flux instantaneously, they affect each other on finite timescales.'

This statement was not written very clearly; it is not an assumption, but a definition. We define two variables v1 and v2 to be quasi-independent if a change in v1 does not imply an instantaneous change in v2, and vice-versa. We will try to clarify this.

The example given in the text is of downwelling SW and ice area. If downwelling SW was to suddenly increase from 50Wm-2 to 100Wm-2, ice area would be unaffected on an instantaneous timescale. If, less realistically, ice area was to suddenly increase from 50% to 100%, downwelling SW would be unaffected on an instantaneous timescale. Clearly these variables are not truly independent – over time, each step change would provoke changes in the local modelled weather that would cause changes in the other model variable. But these changes would not take place immediately. This contrasts to e.g. **up**welling SW and ice area: if ice area suddenly changed from 50% to 100%, the upwelling SW would also change, instantaneously, in response to the surface albedo.

The reviewer's question relates to the relationship between the quasi independent variables vi and the surface heat flux. In the sense described above, a sudden change in any of the vi causes an instantaneous change in surface heat flux. However, a sudden change in surface heat flux does not **cause** a sudden change in the vi (though it might be evidence of such a change having occurred). Rather, it leads to changes to the vi over finite timescales, as the sea ice state, the lowest atmosphere layer, and the top ocean layer respond to the altered surface flux.

In diagnosing surface flux differences due to differences in the considered variables, we are attributing the causes of sea ice growth and melt only in a narrow, proximate sense. We use the quasi-independent framework because it helps disentangle causality: the proximate drivers of surface heat flux differences are those which act at the shortest timescales. As noted above, the ice heat capacity introduces a complicating factor to the 'instantaneous action' view, and we try to explain there why we do not think this invalidates the framework.

We have rephrased the 'quasi-independent' sentence to make clear that this is a definition. In consideration of the issues raised by the internal ice thermodynamics above, we have changed this definition slightly, now requiring only that the variables affect surface flux on a much shorter timescale than that on which they affect each other. In accordance with this, there is a brief discussion for both the melting and freezing season analysis of how well each set of variables fulfils this criterion.

*Equation 3 – superscripts MODEL1 and MODEL2 are not visible.*

Reviewer 1 pointed this out too – this will be amended. This has been amended.

*Line 261: I suggest to write explicitly how the ice volume balance is related to the surface heat flux. I wonder why the ice volume tendency is omitted in the simple model.*

This is a good idea. The relationship between ice volume tendency (ice growth/melt) and surface flux is briefly discussed at the end of Section 3.1, but it would bear stating with an equation at this point in Section 4. The ice volume tendency is in a sense beyond the scope of the simple model itself, whose purpose is to estimate the causes of differences in the surface flux (which we treat as the principal driver of the ice volume tendency).

We have explicitly stated the relationship between ice melt/growth and surface flux with an equation at the beginning of Section 4.

*Equations 5 and 6 – I suggest different letters for the variable a_melt and the area fractions a_i. Maybe, use capital A for the area fractions, otherwise it is confusing.*

This is also a good idea, which Reviewer 1 mentioned. Thank you. All instances of 'a' to describe area fraction have been replaced by 'A'.

*Equation 6 – Fsw-net, t is missing*

This will be corrected. This has been corrected.

*What is the exact definition of a_melt in Section 3 and how is Equation 6 obtained? It is hard to follow.*

a_melt is defined as the fraction of time a grid cell is undergoing surface melting. It is possibly a confusing name – perhaps t_melt would be better – and the definition, which is rather obliquely stated in Section 3, will be restated more clearly here in Section 4.

To show how equation 6 is derived, we will explicitly expand equation 5 in terms of the area and albedo values described in the ensuing paragraph. When monthly mean values are considered, a_melt is functionally equivalent to the average area fraction of ice undergoing surface melting during a month (hence its confusing name). Hence it is closely related to a_meltpond, the average area fraction of a grid cell containing meltpond, in a way that depends upon the meltpond parameterisation. In HadGEM2-ES, a_melt and a_meltpond are related by two constants derived from the albedo parameters, 0.17 and 0.22 (representing the proportion of melting surface covered by meltpond over bare ice and snow respectively). In HadGEM3-GC3.1-LL and UKESM1.0-LL, they are related by the variable a_meltpond / a_melt (again, representing proportion of melting surface covered by meltpond), which can be calculated from model diagnostics.

The description of the melt season analysis has been substantially expanded and clarified. We first express surface albedo as a sum over surface types, and then describe explicitly how this is refined in order to quantify the effects of ice area, snow thickness, snow parameterisation, melting tendency and meltpond fraction: the final net SW equation is then given explicitly (new equation (7)), so that it can be seen how the dependence on net SW on each example variable given is calculated.

As suggested above, we have relabelled A_{melt] as T_{melt} throughout.

*Line 275: We can use this equation – specify which equation*

Equation (5) – this will be stated. This has been stated.

*Obviously, Equation 7 cannot be used for category zero (open water)*

That is true, the derivation is only valid for the ice categories. A different, simpler, equation is used for the open water category, which we will include in our revision. In our revision we have also clarified some notation along these lines. Equation (9) has been relabelled F_{surface-cat}, as it is for individual ice categories only; equation (10) F_{surface_ice}, as it is valid for the ice-covered portion of the grid cell. We have added a new equation (11) for the whole grid cell surface flux, and have described how the open water portion of the surface flux is calculated.

*Line 351: How is it linearized and what is Bup?*

At each model grid cell, Fsfc is linearised about $T_{sfc}$ $(x,t)$, the monthly mean surface temperature at that grid cell averaged between the two models being compared. Bup then represents $\partial F_{sfc}/\partial T_{sfc}\,|_{T_{sfc}}$. In our simple model, we view all components of the surface flux except the upwelling LW flux as being independent of surface temperature – hence $B_{up} = 4\varepsilon\sigma T_{sfc-0}^3$.

The reasoning is described in greater detail in the answer to your next question.

We have expanded this sentence into a paragraph describing the linearisation of the dependence of surface flux on surface temperature in more detail.

*Lines 355-357: First it is stated that Fatmos-ice does not depend on the surface temperature. Next, Fatmos-ice is identified as sum of SW net, LW down and turbulent fluxes. Obviously, turbulent fluxes do depend on surface temperature. It can be argued that LW down also depends on surface temperature on the time scale of the atmospheric boundary layer adjustment (which is not large), because the near-surface air temperature over sea ice is coupled to surface temperature.*

Thank you for raising this. The answer to this question goes to the heart of the issue relating to timescales.

The purpose of this derivation is to separate the effects of snow depth and ice thickness from those of 'external' atmospheric thermal forcing on sea ice growth (or lack of) during the ice freezing season. However, there is no way to define the atmospheric thermal forcing such that it is completely, truly, 'external' (independent of ice thickness), as all elements of the climate system are related. Hence the definition above of 'quasi-independence' – we treat variables as independent if they only affect each other on finite timescales.

The trouble is that this is of course a simplification. Some timescales, while finite, are sufficiently short that it does make more sense for the purposes of what we are trying to do to treat them as being instantaneous. We argue that for a meaningful characterisation of induced surface flux differences, it makes sense to treat both the sensible heat flux, and also the downwelling LW flux, as being independent of surface temperature, and illustrate why we think this is the case with an example.

Consider an ice floe of thickness 1m in typical inverted (cold, clear) Arctic winter conditions. Assume the following conditions, which are fairly realistic: there is negligible SW radiation; the surface skin

temperature is -20C; the air temperature at 2m is -18C; ambient wind conditions are such that there is a substantial sensible heat flux into the surface of 10 Wm-2; the downwelling LW is a fairly typical 190 Wm-2; specific humidity is sufficiently low that latent heat flux can be neglected for the purposes of this example.

From these conditions we can also derive the following: the upwelling LW flux is 228 Wm-2; there is net surface heat flux of -28 Wm-2, indicating cooling and growth of the ice column.

The central question which the ISF analysis is addressing, for each point of model space and time, is: how sensitive is this surface heat flux to differences in ice thickness, and how sensitive to differences in atmospheric thermal forcing? And the question we are trying to address in this example is: how does the treatment of the sensible heat flux affect the answer to the first question?

To judge this, imagine a second ice column of different thickness – let's say 1.5m – to be placed in conditions of 'identical atmospheric forcing'. The key question becomes then how that identical atmospheric forcing is defined, because not all possible definitions give useful characterisations of the 'dependence of surface flux on ice thickness, irrespective of atmospheric forcing'.

Under any meaningful definition of identical atmospheric forcing, a 1.5m ice column is going to transmit heat from the ocean to the atmosphere less efficiently than our original, 1m column, and that because of this the surface of the thicker column is going to converge to a colder surface temperature. Suppose it converges to -24C – this is realistic, as the temperature gradient of the thicker column is then still shallower than that of the thinner column, supporting a weaker surface flux. The upwelling LW flux is a weaker 214 Wm-2; what is the sensible heat flux?

Again, under a meaningful definition of identical atmospheric forcing, we can assume that the ambient wind conditions are identical over our two ice columns; the processes by which ice thickness affects wind speed are too complex, and act over too long a timescale, to be considered as an immediate response to ice thickness. Hence we can assume that the relationship between sensible heat flux, and surface-to-2m air temperature gradient, is the same between our two ice columns. And the 2m air temperature above the 1m ice column was -18C. Does this then mean that we should assume the sensible heat flux to the 1.5m ice column is three times larger than that to the 1m ice column, -30 Wm-2?

No, because the 2m air temperature would in reality respond very quickly to the change in surface temperature, and cannot therefore be considered to be independent of the ice thickness. In other words, the 2m air temperature is not a useful, meaningful diagnostic of atmospheric thermal forcing for these purposes, because it is affected at least as closely by the ice thickness as by the prevailing weather conditions. In a sense, it is almost as much a part of the sea ice system as is the surface temperature. Because of this, we have to view the sensible heat flux as varying extremely weakly with the surface temperature: small perturbations in surface temperature will produce similar perturbations in near-surface air temperature on a very short timescale.

To support this we have examined daily timeseries of surface temperature and sensible heat flux (amongst other atmospheric variables) in our evaluated models, for a few grid cells of the Arctic in winter 1990-91, chosen at random. Figure R1 below, a plot of the trajectory of surface temperature and (upwards positive) sensible heat flux for four grid cells in the Arctic Ocean in Dec 1990-Jan1991, bears out the argument above well: sudden drops in surface temperature are not systematically associated with sudden drops in sensible heat flux (Figure R1). Over the two months, there is no systematic relationship on any timescale; the variability in sensible heat flux is dominated by atmospheric conditions.

[Figure]

*Figure R1. The evolution of surface temperature and sensible heat flux in four grid cells of the Arctic Ocean, Dec 1990-Jan 1991, in UKESM1.0-LL.*

For the purposes of our simple model we feel it is therefore not unreasonable to make the approximation that the sensible heat flux is independent of surface temperature.

Regarding the downwelling LW, the answer is related to the previous one – indeed the adjustment of near-surface air temperature to a change in surface temperature will form part of a large-scale adjustment of the atmospheric boundary layer. The effect of this adjustment on downwelling LW will depend on multiple factors: how deep is the boundary layer, how high is the cloud base, what is the phase mixture in the lowest cloud layer. This includes many processes far too complex to include in our simple model.

However, some general remarks are possible. The LW response to boundary layer temperature changes alone is likely of second-order importance; changes in absolute humidity are likely small on short timescales. It is likely the cloud response that is most important. In a review of mixed-phase clouds in the Arctic boundary layer, Morrison et al. (2012) again present evidence of persistence of two distinct states (mixed-phase clouds and radiatively clear) with occasional rapid transitions between the two; this picture does not support substantial changes in cloud properties in response to short timescale surface temperature changes (in the real world, that is).

In models, the cloud liquid fraction is systematically, usually dramatically, underestimated over the Arctic, and mixed-phase clouds are very rare except in a few models (Pithan et al., 2014). Notably this is the case for all of our own evaluated models – we have evaluated liquid water fraction with respect to MODIS and all of our models have most cloud liquid water fraction in the 0-10% range which is not the case for MODIS (Figure R2, reproduced from West (2021)). We note that this is likely to mean the response of downwelling LW to surface temperature changes is actually lower in our evaluated models than in reality, due to the lower emissivity of ice particles relative to water

droplets. Effectively, it is not obvious to us from the existing evidence that downwelling LW adjustment due to short-timescale response to surface temperature is of a significant magnitude (compared to the upwelling LW response, which is much easier to parameterise). Given this, and the complexity of satisfactorily accounting for this effect in our simple model, we think that our original decision to neglect this effect was in this case justified.

[Figure]

*Figure R2. Histogram of cloud liquid water fraction over the Arctic Ocean modelled by HadGEM2-ES, HadGEM3-GC3.1-LL and UKESM1.0-LL, and measured by MODIS*

As with the ice thermodynamics discussion above, we have condensed the above response into a paragraph in Section 4 so as not to interrupt the logical flow. When we describe the treatment of downwelling LW and turbulent flux as independent of surface temperature, we give a brief justification of this treatment along the lines above.

*Figure 7 and lines 445-450: It should be better explained how the curves in Figure 7 are obtained. Ice melt and ice growth are not described by the model in Section 4. Such terms are simply missing. So it is not clear at all how Figure 7 is obtained.*

But the whole premise of the study is that the surface flux is the principal driver of ice melt and growth – hence we are diagnosing causes of surface flux difference, which we treat as synonymous with ice melt and growth difference. We will try to make this clearer.

We have clarified the surface flux – sea ice melt/growth relationship again with a sentence at the start of Section 5, referencing the new subsection 3.2.

*Line 479: modelled sea ice and growth (??)*

'modelled sea ice melt and growth' – this will be amended. This has been amended.

**References**

Keen, A. and Blockley, E.: Investigating future changes in the volume budget of the Arctic sea ice in a coupled climate model, The Cryosphere, 12, 2855–2868, https://doi.org/10.5194/tc-12-2855-2018, 2018

McPhee, M. G., Kikuchi, T., Morison, J. H., and Stanton, T. P.: Ocean-to-ice heat flux at the North Pole environmental observatory, Geophys. Res. Lett., 30, 2274, https://doi.org/10.1029/2003GL018580, 2003

Morrison, H., de Boer, G., Feingold, G. et al. Resilience of persistent Arctic mixed-phase clouds. Nature Geosci 5, 11–17 (2012). https://doi.org/10.1038/ngeo1332

Perovich, D. K., Richter-Menge, J. A., Jones, K. F., and Light,B.: Sunlight, water, and ice: Extreme Arctic sea ice melt during the summer of 2007, Geophys. Res. Lett., 35, L11501,https://doi.org/10.1029/2008GL034007, 2008

Pithan, F., Medeiros, B., and Mauritsen, T.: Mixed-phase clouds cause climate model biases in Arctic wintertime temperature inversions, Clim. Dynam., 43, 289–303, https://doi.org/10.1007/s00382-013-1964-9, 2014

Serreze, M. C., Barrett, A. P., Slater, A. G., Steele, M., Zhang, J., and Trenberth, K. E. (2007), The large-scale energy budget of the Arctic, J. Geophys. Res., 112, D11122, doi:10.1029/2006JD008230

Steele, M., Zhang, J., and Ermold, W.: Mechanisms of summer Arctic Ocean warming, J. Geophys. Res.-Oceans, 115, C11004, https://doi.org/10.1029/2009JC005849, 2010

West, A., New methods for evaluating sea ice in climate models based on energy budgets, PhD thesis, https://ore.exeter.ac.uk/repository/handle/10871/127762 2021

**Appendix: how to characterise timescales of ice temperature response to changes in surface forcing**

**Statement of the problem**

Consider an ice column of thickness $h$, with base temperature $T(t, -h) = T_{bot}$ $\forall t$. We ignore turbulent and shortwave fluxes, hence the principal forcing on the ice is the downwelling LW forcing $F_{LW\downarrow}(t)$, and the surface flux

$$F_{sfc} = F_{LW\downarrow} - \varepsilon\sigma T_{sfc}^4 \qquad\qquad (1)$$

where $\epsilon$, $\sigma$ and $T_{sfc}$ denote emissivity, the Stefan-Boltzmann constant and surface temperature respectively.

By flux continuity also, $F_{sfc} = k_I \left.\frac{\partial T}{\partial z}\right|_{z=0}$ (we assume uniform, constant, ice conductivity $k_I$ throughout the ice column.

We assume that for all time t<0, the ice column is in thermodynamic equilibrium, forced by a constant downwelling LW flux $F_{LW\downarrow}^1$, with a linear temperature profile $T(z,t) = T_{sfc} + \frac{z(T_{bot} - T_{sfc})}{h}$, with $T_{sfc}$ chosen to satisfy flux continuity.

At time t=0, the downwelling longwave flux abruptly changes from $F_{LW\downarrow}^1$ to $F_{LW\downarrow}^2$. As $t \to \infty$ the temperature profile will approach a new straight line. The problem then is to determine the timescale in which the temperature profile decays to the new equilibrium, and how this timescale depends on $h$.

**Solution**

To simplify the maths we carry out a change of vertical coordinate: $Z = z + h$. Hence the bottom boundary condition becomes $T(t, 0) = T_{bot}$.

The evolution of the ice temperature is described by the standard heat equation

$$\rho c_p \frac{\partial T}{\partial t} = k_I \frac{\partial^2 T}{\partial Z^2} \qquad (2)$$

solutions to which are sums of terms of the form

$$A e^{-\lambda \alpha t} sin\left(Z\sqrt{\lambda}\right) \qquad (3)$$

for $\lambda > 0$, where $\alpha = k_I/\rho c_p$.

This much is standard theory; the unique contribution of the ice column problem is the top boundary condition which restricts λ to a particular discrete set:

$$k_I \frac{\partial T}{\partial Z}\bigg|_{Z=h} = F_{LW\downarrow}^2 - \varepsilon \sigma T^4(t, h) \qquad (4)$$

We linearise this and substitute in $T = A e^{-\lambda \alpha t} sin\left(Z\sqrt{\lambda}\right)$ to find

$$-k_I \sqrt{\lambda} \, cos\left(h\sqrt{\lambda}\right) = B \, sin\left(h\sqrt{\lambda}\right) \qquad (5)$$

where $B$ is the rate of dependence of outgoing longwave radiation on surface temperature.

Roots to this equation can be found numerically, and indeed tend to $(n + 1/2)\pi$ as $n \to \infty$. However we are interested in the long timescale response of T, and are hence interested in the first, smallest root of equation (5) as this will produce the slowest-decaying harmonic. Solving numerically for the root in the interval $(\pi/2, \pi)$ we find the following relationship between $h$ and the decay timescale (Figure R3):

[Figure]

*Figure R3. The relationship between ice thickness and time taken to relax towards a new temperature profile*

These results also form the basis of Table R1.

**Complete list of changes to the manuscript**

In the list below, all line numbers refer to the tracked changes version of the manuscript. Changes which relate solely to renumbering of figures, equations or sections following insertion of new elements are not included in this list.

**Line 2:** co-authors' first names amended by request

**Line 10:** 'A system of simple models' amended to 'An energy balance approach' for clarity, and for consistency with a revised explanation of the approach in Section 3.

**Line 20:** '106' corrected to '$10^6$'

**Line 85:** Sea ice components of the different models clarified (Reviewer 1 suggestion)

**Line 86:** Wording 'the CMIP6 models' used throughout the study defined

**Line 113:** extraneous 'and' removed (R1)

**Line 167:** new Section 3.2 inserted, with new Figure 2, justifying treatment of ice growth/melt as consequence of surface flux

**Line 233:** sentence amended as suggested by Reviewer 1

**Line 239:** surface albedo parameterisation discussed in more detail, as requested by Reviewer 2

**Line 250:** new sea ice area fraction row on Figure 4 (previously 3) described.

**Line 261:** New Figure 4 (previously 3) with sea ice area fraction row inserted at top, and observational reference column inserted on right; caption updated accordingly.

**Line 269:** All instances of $a_i$ to denote area fraction replaced with $A_i$, for greater distinction from albedo $\alpha_i$ (R1 & R2)

**Line 273:** variables defined (R2)

**Line 278:** 'ice area' corrected to 'snow area' (R2)

**Line 285:** HadGEM2-ES meltpond parameterisation described in greater detail (R2)

**Line 296**: model biases wrt NSIDC melting tendency reference dataset described (R1)

**Line 306:** model biases wrt MODIS meltpond fraction reference dataset described (R1)

**Line 313:** Paragraph & new Figure 5 inserted to estimate surface albedo difference caused by each variable (R1 suggestion)

**Line 335:** new paragraph to link from here to Section 4.

**Line 347:** Paragraph amended and expanded, to make explicit the link between ice growth/melt and surface flux, and to link back to previous section.

**Line 353:** 'Quasi-independent' definition generalised (action does not have to be instantaneous) and clarified (R2)

**Line 362:** Link between quasi-independence and causality discussed in greater detail

**Line 366:** again the link between surface flux and ice growth/melt is made

**Line 368:** introductory paragraph to the derivation is slightly altered, as the derivation is now described in more detail than in the previous revision.

**Line 375:** The melt season analysis is now described in more detail. Albedo is explicitly split into components. We describe how the equation is then rearranged so that the surface flux is expressed in terms of the variables examined in Section 3.

**Line 406:** Albedo parameters are defined

**Line 415:** previous numerical example deleted as we felt it was no longer needed

**Line 420:** derivation of Equation (8), previously (6), is now clearer (R2)

**Line 436:** melt season analysis is now limited to the first ensemble member only when HadGEM2-ES is the reference model (R1 suggestion)

**Line 458:** spatial patterns of melt season ISF differences described (R1)

**Line 470:** differences between snow area and ice area components discussed (R1); these are different to those shown in the original version due to the use of the full ensemble for the UKESM1.0-LL and HadGEM3-GC3.1-LL comparison.

**Line 480:** new Figure 6 (previously 4) inserted with ensemble mean & standard deviation shown in second panel, and with contrasting linestyles (R1) suggestion. Caption amended accordingly.

**Line 492:** For consistency, all instances of $x_{surface}$ to denote surface variables have been renamed $x_{sfc}$

**Line 499:** Linearisation of surface flux dependence on surface temperature is described more explicitly (R2)

**Line 503:** The surface types for which each equation is valid have been clarified. This equation (11, previously 9), is valid only for a single ice category.

**Line 506:** This equation (12, previously 10) is valid for the ice-covered surface of a grid cell

**Line 513:** Paragraph justifies treatment of downwelling LW and turbulent fluxes as independent of surface temperature (R2)

**Line 519:** New equation giving surface flux for the entire grid cell, describing calculation of open water portion (R2)

**Line 526:** Quasi-independence of the ice thickness and atmospheric variables is demonstrated, paraphrasing ice thermodynamics discussion in reply to Reviewer 2

**Line 555:** spatial patterns of freezing season ISF differences described (R1)

**Line 563:** rewording carried out according to Reviewer 1 suggestion.

**Line 570:** Figure 7 (previously 5) amended with contrasting linestyles (R1)

**Line 578:** inclusion of ice export on Figure 8 (previously Figure 6) described

**Line 590:** role of ice export term discussed

**Line 609:** new Figure 8 (previously Figure 6) inserted, with ice export term shown, and contrasting linestyles / markers.

**Line 618:** surface flux – ice growth/melt link again emphasised

**Line 673:** 'sea ice and growth' replaced with 'sea ice melt and growth' (R2)

**Line 714:** acknowledgements updated to reflect reviewers' contribution

**Line 719:** all new references used in the text inserted.